# Conducting Polymers-Based Gas Sensors: Principles, Materials, and Applications

**DOI:** 10.3390/s25092724

**Published:** 2025-04-25

**Authors:** Rongqing Dong, Mingna Yang, Yinxiu Zuo, Lishan Liang, Huakun Xing, Xuemin Duan, Shuai Chen

**Affiliations:** 1Jiangxi Provincial Engineering Research Center for Waterborne Coatings, School of Chemistry and Chemical Engineering, Jiangxi Science & Technology Normal University, Nanchang 330013, China; drongqing@163.com (R.D.); yangmingna1023@163.com (M.Y.); lianglishan09@163.com (L.L.); 2Jiangxi Provincial Key Laboratory of Flexible Electronics, Jiangxi Science & Technology Normal University, Nanchang 330013, China; xhk1235@126.com (H.X.); duanxm@jxstnu.edu.cn (X.D.); 3Institute of Energy Materials and Nanotechnology, School of Civil Engineering and Architecture, Nanchang Jiaotong Institute, Nanchang 330100, China; 4School of Chemistry and Chemical Engineering, Jinggangshan University, Ji’an 343009, China

**Keywords:** conducting polymer, gas sensor, sensing mechanism, environmental monitoring

## Abstract

Conducting polymers (CPs) have emerged as promising materials for gas sensors due to their organic nature coupled with unique and versatile optical, electrical, chemical, and electrochemical properties. This review provides a comprehensive overview of the latest developments in conducting polymer-based gas sensors. First, the fundamental gas sensing mechanisms in CPs-based sensors are elucidated, covering diverse transduction modes including electrochemical, chemiresistive, optical, piezoelectric, and field-effect transistor-based sensing. Next, the various types of conducting polymers employed in gas sensors, such as polypyrrole, polyaniline, polythiophene, and their composites are introduced, with emphasis on their synthesis methods, structural characteristics, and gas sensing response properties. Finally, the wide range of applications of these sensors is discussed, spanning industrial process control, environmental monitoring, food safety, biomedical diagnosis, and other fields, as well as existing issues such as long-term stability and humidity interference, and a summary of the biocompatibility and regulatory standards of these conductive polymers is provided. By integrating insights from sensing mechanisms, materials, and applications, this review offers a holistic understanding of CPs-based gas sensors. It also highlights future research directions, including device miniaturization, AI-assisted gas identification, multifunctional integrated sensing systems, wearable and flexible sensor platforms, and enhanced sensitivity, selectivity, and on-site detection capabilities.

## 1. Introduction

Environmental pollution, public health concerns, and industrial safety needs have created an urgent demand for advanced gas-sensing technologies. Rapid industrialization and urbanization have led to significant harmful gas emissions, with pollutants such as sulfur dioxide (SO_2_) [1], nitrogen oxides (NO_X_) [2], and volatile organic compounds (VOCs) [3] threatening air quality. Accurate detection of these hazardous gases is essential for evaluating air quality and formulating effective environmental policies to safeguard the atmosphere critical to human health. In industrial settings, real-time monitoring of gas concentrations is equally crucial for safety and process control. It can prevent accidents (e.g., explosions or toxic exposure) and help maintain product quality and production efficiency. For instance, industries such as chemical manufacturing [4] and food processing [5] rely on precise gas detection to ensure safe operations and consistent product quality. In the medical field, analysis of specific components in exhaled breath provides a non-invasive approach to early disease diagnosis and personalized health monitoring [6]. For example, elevated acetone levels in the breath of diabetic patients serve as an important biomarker of their condition and treatment efficacy [7]. These diverse applications underscore that gas detection technology has become indispensable in modern society.

However, while effective for certain purposes, conventional gas detection methods have significant limitations [8]. For instance, gas chromatography-mass spectrometry (GC-MS) offers high sensitivity and resolution for identifying and quantifying gases in complex mixtures, but it requires bulky, expensive equipment and skilled operators. Its complex, time-consuming procedures make GC-MS impractical for rapid on-site or real-time monitoring [9,10,11]. Similarly, optical spectroscopic methods (such as infrared sensing) enable non-contact and fast detection, yet often suffer from low sensitivity for certain gases and depend on high-cost instrumentation [12]. These drawbacks have driven researchers to explore new sensing materials and techniques to meet the growing demand for gas sensors with higher precision, sensitivity, and selectivity under practical conditions.

To overcome these challenges, conducting polymers (CPs) have emerged as promising gas sensor materials offering unique advantages. CPs are organic polymers with π-conjugated backbones that endow them with intrinsic electrical conductivity. Moreover, their electrical, optical, and chemical properties can be readily tuned through chemical modification or doping [13], allowing for sensor designs tailored to specific target gases. Metal oxides and metal-organic frameworks (MOFs) have shown excellent adsorption and selective detection performances toward gas sensing, but they often need to work under high-temperature conditions [14,15]. In contrast, the great advantages of CPs in this field are that they can work at room temperature, quickly respond to gases through intermolecular interactions, enable solution processing, have low power consumption, as well as be suitable for flexible integration and portable monitoring [16].

Several CPs materials have been extensively studied in gas sensors, notably polyaniline (PANi) [16], polypyrrole (PPy) [17], polythiophene (PTh) [18], poly(3,4-ethylenedioxythiophene) (PEDOT) [19], and its derivatives PEDOT:PSS [20]. Each of these CPs provides distinct characteristics beneficial for gas detection. For example, PANI features excellent environmental stability and a widely tunable conductivity, which has been leveraged to detect gases such as ammonia and nitrogen dioxide [16]. PPy is easily synthesized and forms uniform conductive films with high electrochemical activity, making it effective for sensing various VOCs [17]. PTh and its derivatives offer unique electronic and optical properties that can be exploited to enhance selectivity toward specific gas molecules [18]. PEDOT is known for its high conductivity and stability [19], while PEDOT:PSS combines the conductivity of PEDOT with the water dispersibility of PSS, enabling solution-processed, flexible sensor devices [20].

Thanks to these advantages, CPs-based gas sensors have demonstrated promising performance across a wide range of applications, including environmental monitoring [21], industrial process control [22], medical diagnostics [23], and food safety assurance [24]. Numerous studies have validated the utility of CPs sensors in such areas; however, despite the considerable progress, several challenges remain. In particular, further improvements are needed in detection sensitivity and selectivity, expansion of the range of detectable gases, and enhancement of long-term stability and reliability. In light of both the achievements and the remaining challenges in this field, this review aims to provide a comprehensive overview of CPs-based gas sensor technology. We discuss the fundamental gas sensing mechanisms and the diverse CPs materials employed, summarize recent advances in various application domains, and finally outline current challenges and future research prospects for CPs-based gas sensors in this rapidly evolving field.

## 2. Conductive Polymer-Based Gas Phase Sensors

### 2.1. Detection Principle

#### 2.1.1. Electrochemical Sensing

Electrochemical sensing utilizes redox reactions occurring at the electrode–electrolyte interface to detect electrochemically active gases, converting chemical information into measurable electrical signals [25]. Electrochemical gas sensors are predominantly classified into potentiometric, conductometric, and amperometric methods. The potentiometric method operates based on the reaction of the target gas at the working electrode surface, generating a potential difference between the working and reference electrodes, which varies with changes in gas concentration [26]. The conductometric method relies primarily on changes in the conductivity of sensitive materials upon gas interaction to detect the analyte. Among these approaches, the amperometric method has become the most widely utilized due to its high sensitivity and strong quantitative detection capabilities. A typical amperometric gas sensor comprises a working electrode (WE), a counter electrode (CE), and a reference electrode (RE). The WE is the principal site where oxidation or reduction of the target gas occurs, initiating electron transfer and generating a current or potential shift directly correlated to analyte concentration [27]. When detecting oxidizing gases (e.g., NO_X_, O_3_), the target molecule accepts electrons at the cathodic interface, generating a measurable cathodic current [28]. Conversely, reducing gases (e.g., H_2_S, NH_3_, CO) donate electrons upon oxidation at the anodic interface, producing a characteristic signal. The magnitude of the response follows Nernst’s equation ($E=E^{0}+\frac{RT}{nF}\ln\frac{\left[O_{x}\right]}{\left[Red\right]}$) and Faraday’s law ($m=\frac{M}{nF}Q=\frac{M}{nF}It$) providing quantitative information on the gas concentration [25,26].

CPs are ideal candidates for electrochemical gas sensors due to their intrinsic redox activity, high electrical conductivity, and tunable surface chemistry [29]. Their π-conjugated backbones facilitate efficient charge transport, and their functional groups or dopants can be tailored to enhance selectivity toward specific analytes. For example, Serafini et al. [30] developed a wearable electrochemical ammonia gas sensor with core components including PEDOT, electrochemically deposited iridium oxide particles, and a hydrogel membrane. Upon contact with ammonia gas, a dissociation equilibrium within the hydrogel releases OH^−^ ions, increasing the hydrogel’s pH (Figure 1a,b). This pH shift disrupts the redox equilibrium of iridium oxide (IrOx), facilitating electron injection into PEDOT:PSS, consequently reducing its conductivity. Under an applied voltage between two electrodes, a measurable current decrease occurs. Monitoring this current reduction allows for accurate ammonia gas detection. Following ammonia introduction, a rapid current drop illustrates the efficiency of the detection mechanism.

Despite their benefits, electrochemical gas sensors are primarily effective for gases possessing intrinsic redox activity, such as carbon monoxide, hydrogen sulfide, and ammonia. In contrast, inert gases like chlorine or nitrogen require alternative detection approaches or indirect redox mechanisms [31]. Thus, continued optimization of electrode materials, electrolyte compositions, and CPs electrode interfaces remains essential for broadening detection capabilities and enhancing sensor performance.

**Figure 1 sensors-25-02724-f001:**
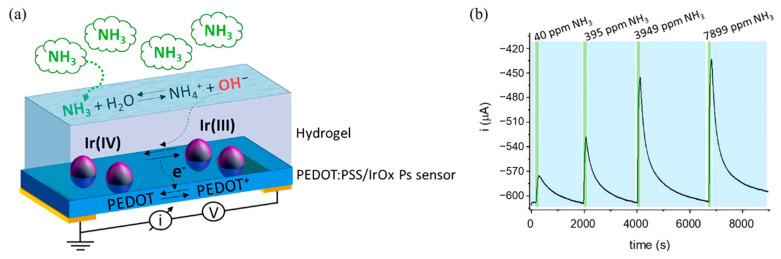
(**a**) Schematic of the gas sensor working principle; (**b**) current vs. time response of the NH_3_ sensor [30].

#### 2.1.2. Chemiresistive Sensing

Chemiresistive sensing is based on detecting changes in the electrical resistance of sensing materials upon interaction with target gases [32]. Specifically, gas molecules adsorb onto the sensor’s active layer, triggering physical or chemical interactions that alter its electronic properties, such as charge carrier concentration, mobility, or barrier height, thereby producing measurable changes in resistance [33]. CPs are particularly suitable for chemiresistive sensors due to their tunable electrical properties, ease of chemical modification, high sensitivity, and rapid response capability. When a target gas interacts with CPs-based sensing materials, it can function as either an electron donor or acceptor. For example, electron-donating gases (e.g., NH_3_, acetone) increase electron density in p-type CPs, thereby reducing hole concentration and consequently increasing the polymer’s resistance [34]. Conversely, electron-accepting gases (e.g., NO_2_) withdraw electrons, increasing hole concentration and decreasing resistance.

A representative example is the SnO_2_/PTh nanocomposite sensor developed by Beniwal et al. [35] for acetone detection. Upon exposure to acetone gas, acetone molecules act as electron donors, transferring electrons to polythiophene (PTh, a typical p-type semiconductor) (Figure 2a). This electron transfer decreases the concentration of holes (majority charge carriers) in PTh, significantly increasing its electrical resistance. Since PTh coats the SnO_2_ nanoparticles, this resistance change directly influences the composite’s overall conductivity, enabling precise acetone detection through resistance measurements. In another study, Chaudhary et al. [36] developed a chemiresistive sensor by combining polythiophene (PTh) with citric acid-functionalized cadmium sulfide quantum dots (CdS QDs) for ammonia sensing. Ammonia molecules donate lone-pair electrons upon adsorption, interacting with the polymer’s polarons. This interaction disrupts the charge balance and restricts carrier mobility within the PTh/CdS composite, resulting in a pronounced decrease in electrical resistance [37]. Notably, this sensor demonstrated a rapid response (under 1 s) and recovery (4–8 s), with a clear linear dependence on ammonia concentration, highlighting the exceptional suitability of PTh-based composites for chemiresistive ammonia detection.

Despite significant advancements, challenges remain for chemiresistive sensors, including cross-sensitivity to humidity, baseline drift, and long-term stability [38]. Current research efforts therefore emphasize enhancing selectivity through molecular design and composite engineering and developing strategies to mitigate environmental interference, further advancing CPs-based chemiresistive sensors.

#### 2.1.3. Piezoelectric Sensing

Piezoelectric sensing involves converting mechanical deformation induced by gas adsorption into measurable electrical signals, utilizing materials exhibiting the piezoelectric effect [39]. This sensing mechanism relies on changes in mass, viscoelastic properties, or mechanical stress within piezoelectric materials upon interaction with target gases [40]. CPs and their composites are increasingly employed in piezoelectric gas sensors due to their flexibility, ease of modification, and tunable piezoelectric properties. Typically, piezoelectric gas sensors utilize quartz crystal micro-balance (QCM) or polymer-based piezoelectric substrates coated with CPs-sensitive layers. Upon gas adsorption, mass loading or interfacial interactions between gas molecules and CPs coating cause frequency shifts or mechanical stress changes, generating electrical signals proportionate to the gas concentration [41]. The advantage of using CPs coatings includes their customizable affinity toward specific gas molecules and improved mechanical flexibility, crucial for wearable and flexible electronics applications.

Adjaoud et al. [42] developed a flexible piezoelectric sensor utilizing ionic polymer-polymer composites (IP_2_Cs), incorporating PEDOT:PSS-modified electrodes. Mechanical stimulation induced ion migration within the composite, generating measurable voltage signals due to asymmetric charge distribution (Figure 2b,c). The integration of PEDOT:PSS facilitated improved electrical conductivity and sensitivity, demonstrating significant potential for wearable gas sensing applications. Furthermore, combining CPs with inorganic piezoelectric materials, such as carbon nanotubes (CNTs) or graphene oxide (GO), can further enhance sensing performance. For instance, Pasupuleti et al. [43] prepared a NO_2_ sensor based on GO-PEDOT:PSS nanocomposites, which showed good stability, enhanced sensitivity, and improved response toward NO_2_.

Despite promising outcomes, piezoelectric CPs-based gas sensors continue to face challenges, such as sensitivity to humidity and issues related to long-term stability [44]. Current research aims to optimize composite formulations, refine device architectures, and mitigate environmental interferences to facilitate broader practical applications.

**Figure 2 sensors-25-02724-f002:**
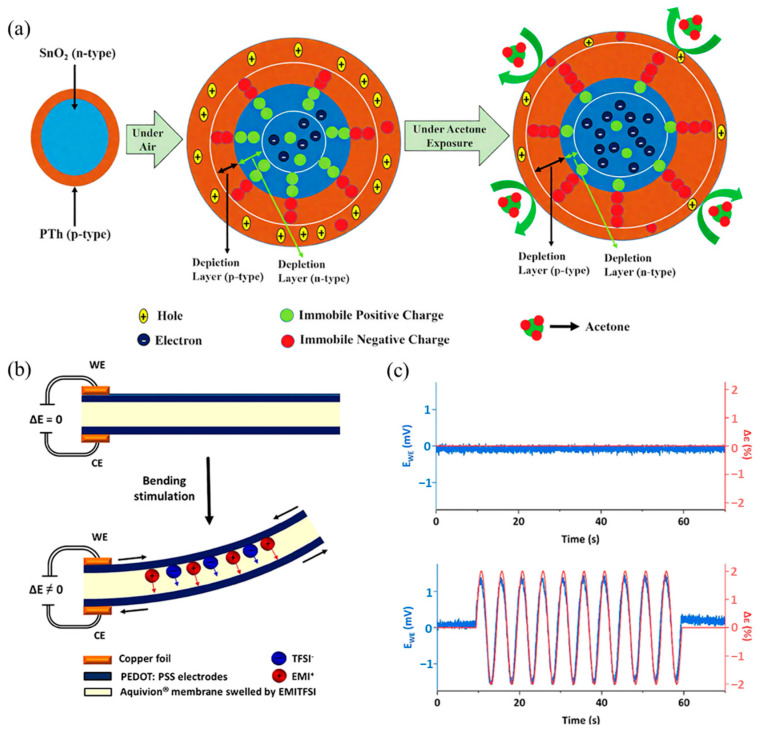
(**a**) Schematic of sensing mechanism of the SnO_2_/PTh nanocomposite toward acetone detection [35]; representation of the piezoionic effect. (**b**) Schematic illustration of the tri-layer before and after mechanical stimulation and (**c**) electrical response (voltage output) of IP_2_Cs sensor before and after mechanical stimulation [42].

#### 2.1.4. Mechanical Sensing

Mechanical sensing transforms mechanical stimuli, such as strain, stress, pressure, or deformation, into quantifiable electrical signals by altering conductive networks or electron tunneling paths within conductive materials [45]. CPs have emerged as promising materials for mechanical sensing owing to their inherent flexibility, tunable mechanical and electrical properties, ease of processing, and capability to integrate with various substrates, making them highly suitable for wearable and flexible sensing applications.

CPs-based mechanical sensors typically operate through mechanisms such as changes in tunneling resistance, contact separation, and crack propagation [46]. Under mechanical deformation, the conductive pathways within CPs composites experience structural modifications, altering inter-particle distances and particle alignment, or causing micro-crack formation, significantly impacting electrical conductivity [47]. For instance, Mallya et al. [48] described poly(DTCPA-co-BHTBT)-CB composites used for sensing toluene vapor. The absorption of toluene vapor causes polymer swelling, akin to mechanical deformation effects on CPs composites. This swelling increases the spacing between carbon black particles, disrupting the carbon black network structure and resulting in conductivity changes that enable toluene vapor detection. This exemplifies how CPs-based materials exploit structural changes to induce conductivity variations, achieving sensing capabilities in different scenarios.

Despite their promising characteristics, CPs-based mechanical sensors still encounter challenges regarding sensitivity, stability, mechanical robustness, and resilience to environmental factors [49]. Future research efforts should focus on optimizing composite formulations, engineering hierarchical structures to improve sensor durability, and addressing baseline drift and sensitivity recovery issues for sustained practical applications.

#### 2.1.5. Optical Sensing

Optical sensing detects target gases by monitoring changes in optical properties, such as absorbance, fluorescence intensity, refractive index, and reflectivity, caused by interactions between gas molecules and sensitive materials [50]. In recent years, in addition to traditional fluorescence and absorption sensors, emerging technical directions in this area such as plasmon-enhanced CPs fluorophores [51,52], waveguide-integrated CPs interferometers [53], and cavity-coupled absorption sensors [54] have also been developed. CPs hold particular promise for optical gas sensing due to their intrinsic optical activity, tunable molecular structures, and ease of functionalization, allowing precise modulation of optical responses upon gas exposure.

Typical CP-based optical gas sensors include fluorescence-based [55], absorption-based [56], and refractive index-based [57] sensors. In fluorescence sensing, gas molecules interact with CPs films, resulting in fluorescence quenching or enhancement. Lee et al. [58] developed an optical gas sensor by integrating CMOS-MEMS technology with CP-based fluorescent sensing materials (Figure 3a). This sensor employed fluorescence quenching, where blue LEDs excite the sensing material, producing fluorescence, and exposure to the target gas reduces fluorescence intensity through molecular interactions, resulting in a measurable decrease in photocurrent [59,60]. This technique demonstrates rapid response and high sensitivity, making it suitable for on-site gas detection. Additionally, CP-based absorption optical sensors detect changes in absorption spectra caused by gas adsorption. Liu et al. [61] developed an ethanol gas sensor using polypyrrole (PPy)-modified plastic optical fibers (POFs). Upon ethanol exposure, PPy interacts with ethanol molecules, altering its refractive index and absorption characteristics, subsequently affecting the transmitted optical signal intensity. Such sensors exhibit fast response times and can be easily integrated into compact, portable devices, ideal for practical applications.

Despite significant progress, CP-based optical sensors still face challenges concerning sensitivity limits, humidity interference, and long-term optical stability. Ongoing research focuses on improving the design of sensing materials, optimizing optical signal processing techniques, and developing robust integration strategies to enhance sensor performance for real-world applications [62].

#### 2.1.6. Field-Effect Transistor (FET) Sensing

Field-effect transistor (FET) gas sensors operate by detecting changes in the electrical conductivity of the transistor channel caused by interactions between gas molecules and the active sensing layer [63]. CPs are particularly attractive as channel materials in FET-based gas sensors due to their high electrical conductivity, ease of processing, chemical tunability, and potential for low-cost, flexible, and wearable sensing applications. A typical CPs-based FET gas sensor comprises three electrodes: a source, a drain, and a gate [64]. In operation, gas molecules adsorb onto the CPs channel layer. This adsorption induces charge transfer or polarization effects that alter the charge carrier density and mobility within the polymeric semiconductor channel [65]. These changes modulate the channel current, thereby enabling sensitive and selective detection of target gases. For example, FETs employing polythiophene derivatives or polyaniline as the channel material have demonstrated excellent performance for gas detection. Amer et al. [66] fabricated an ammonia gas sensor based on an organic field-effect transistor (OFET) architecture using polyaniline (PANi) and its derivatives as the active channel materials. When this sensor is exposed to ammonia (NH_3_) gas, ammonia molecules react with protonated sites in the polymer to form ammonium ions (NH^4+^), releasing electrons from their lone pairs in the process (Figure 3b). For instance, in a PANi:DBSA-doped device, this reaction leads to a decrease in hole density in the polymer, which in turn significantly reduces the transistor’s channel current. By monitoring the change in channel current, the ammonia concentration can thus be accurately determined at room temperature. Similarly, PEDOT:PSS, well-known for its excellent conductivity and environmental stability, has also been widely used as an active channel material in FET gas sensors. Owing to its strong interactions with polar analytes such as NO_2_, SO_2_, and volatile organic compounds (VOCs), PEDOT:PSS-based FET sensors exhibit high sensitivity, rapid response, and excellent selectivity [67]. Moreover, the solution processability of PEDOT:PSS facilitates the integration of these sensors into flexible and wearable sensing devices.

Despite these promising advances, several challenges remain, particularly related to sensor stability, environmental interference, and baseline drift [68]. Ongoing research efforts are focusing on developing composite CPs materials, implementing surface modifications, and designing advanced device architectures to enhance sensing performance, improve selectivity, and extend the operational stability of CPs-based FET gas sensors.

**Figure 3 sensors-25-02724-f003:**
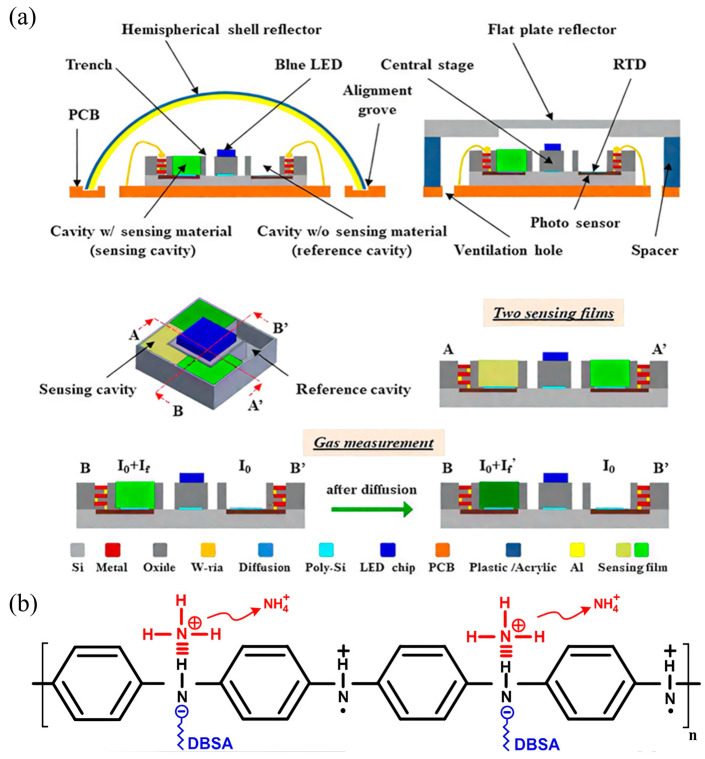
(**a**) Schematic of the presented gas sensor design principle and cross-section [58]; (**b**) schematic diagram of the sensing mechanism of PANi:DBSA [66].

### 2.2. Sensitive Material

#### 2.2.1. PANi

Polyaniline (PANi) has been extensively studied as a CP for gas sensing owing to its distinctive properties, such as facile synthesis, environmental stability, reversible doping/dedoping behavior, and tunable conductivity [69,70,71]. PANi can be synthesized by either chemical or electrochemical polymerization [72]. In chemical synthesis, PANi is typically produced via oxidative polymerization of aniline monomers using oxidants like ammonium persulfate. In contrast, electrochemical polymerization enables the direct formation of PANi layers on conductive substrates under controlled potentials, which facilitates sensor fabrication. PANi exhibits notable advantages for gas sensing applications, most prominently a reversible redox behavior that allows efficient electron transfer when interacting with various analyte gases. Yuan et al. [73] demonstrated highly sensitive ammonia sensors by combining protic acid-doped PANi (PA-PANi) with graphene oxide (GO) and reduced graphene oxide (rGO). Their PA-PANi/GO/rGO composite sensor showed a 262.5% greater response at 25 ppm NH_3_, along with significantly faster response and recovery times, compared to a sensor based on pure PANi, (Figure 4a).

However, PANi also has inherent limitations, including relatively low intrinsic conductivity, poor processability, and limited solubility that constrain its utility in large-scale sensor applications [74]. To address these issues, researchers have extensively explored composite approaches. Bibi et al. [16] enhanced PANi’s H_2_S sensing performance by incorporating carbon aerogel (CA) in an interdigital electrode structure, which significantly improved sensitivity toward hydrogen sulfide. In this configuration, the porous CA framework provides abundant active sites for gas adsorption, thereby enabling rapid and sensitive detection of ultralow H_2_S concentrations. Flexible sensor applications have also benefited from PANi composites. For instance, Wan et al. [75] developed a flexible NH_3_ sensor by depositing a PANi–carbon nanotube (CNT) composite onto a polyethylene terephthalate (PET) substrate. The resulting sensor exhibited remarkable sensitivity with a detection limit as low as 1 ppm, making it suitable for applications such as breath analysis and food safety monitoring. Similarly, Zhuang et al. [76] combined PANi with multi-walled carbon nanotubes (MWCNTs), achieving a detection limit of 0.3 ppm for NH_3_ as well as high selectivity and excellent stability under varying humidity and mechanical stress conditions. Furthermore, PANi-based sensors have shown promise in hydrogen detection, which is critical for safety in hydrogen energy applications. Askar et al. [77] investigated various nanostructured PANi materials and found that PANi hollow nanotubes exhibited outstanding hydrogen-sensing capabilities, including a detection limit as low as 1 ppm, a high sensitivity of about 29%, and rapid response and recovery times (15 s and 17 s, respectively) (Figure 4b,c).

**Figure 4 sensors-25-02724-f004:**
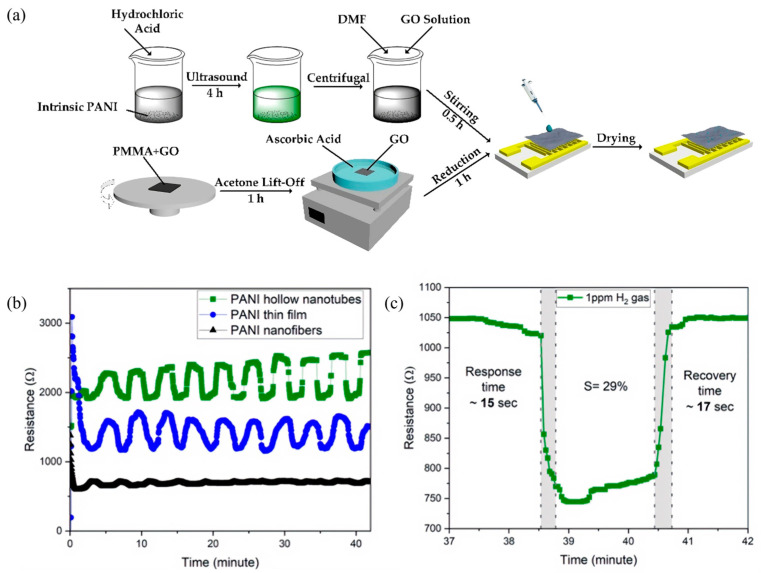
(**a**) Schematic of the device fabrication process [73]; (**b**) comparison of responses in different PANi at 1 ppm H_2_ detection; (**c**) response time of hollow PANi nanotubes sensor at 1 ppm H_2_ gas [77].

Overall, although PANi-based sensors hold considerable promise for gas sensing, challenges remain in improving their intrinsic conductivity, long-term stability, and environmental resilience. Future research should focus on optimizing composite formulations, refining molecular designs, and developing advanced sensor architectures to overcome these limitations and broaden the scope of practical applications.

#### 2.2.2. PPy

Polypyrrole (PPy) is a highly promising CP widely utilized in gas sensing applications due to its intrinsic conductivity, good environmental stability, ease of synthesis, and tunable electrochemical properties [78,79,80]. PPy can be synthesized via chemical or electrochemical polymerization of pyrrole monomers [81]. Chemical polymerization typically involves pyrrole monomers and oxidants such as ammonium persulfate, while electrochemical polymerization allows precise control of PPy morphology and thickness by adjusting deposition parameters [82].

PPy-based sensors demonstrate notable advantages, including low fabrication costs, straightforward preparation methods, biocompatibility, and suitability for flexible and wearable devices [83]. For instance, Gai et al. [84] synthesized PPy-tetra-β-carboxyl cobalt phthalocyanine tetrasodium salt (PPy-TcCoPc) nanorod composites via one-step in situ polymerization. The synergistic interaction between PPy and TcCoPc significantly enhanced ammonia detection performance, achieving high sensitivity (response of 49.3% at 50 ppm NH_3_), rapid response (8.1 s), and excellent selectivity, stability, and humidity resistance (Figure 5a). However, pure PPy often exhibits limitations, including relatively low sensitivity, inadequate selectivity, and poor recovery due to its disordered aggregation structure [85]. To address these issues, researchers have combined PPy with various nanomaterials to enhance sensing performance. Santos-Ceballos et al. [86] developed PPy@laser-induced graphene (LIG) nanocomposite sensors through electrochemical polymerization, achieving remarkable ammonia sensitivity with a detection limit of 1 ppm and superior repeatability due to the synergistic interactions between PPy and LIG. Furthermore, PPy-based composites have shown promising results in detecting other hazardous gases such as hydrogen sulfide (H_2_S) [87]. Al-Sabagh et al. [88] fabricated PPy composites incorporating CuO and SnO_2_ nanoparticles, demonstrating significant improvements in H_2_S detection performance attributed to the nanoparticles’ large surface areas and enhanced electrical properties.

Future research directions for PPy-based gas sensors include optimizing composite formulations, improving sensor architectures, and enhancing environmental resilience and stability. Ongoing studies focus on incorporating novel materials with PPy to further refine sensor sensitivity, selectivity, and long-term reliability, promoting broader practical applications across industrial, environmental, and medical fields [89,90].

#### 2.2.3. PTh

Polythiophene (PTh) is a CP known for its excellent electrical conductivity, chemical stability, and easily tunable molecular structure, making it highly suitable for gas sensing applications [91,92]. The backbone of PTh, comprising conjugated thiophene units, facilitates efficient electron transport and strong interactions with gas analytes [93]. These characteristics allow PTh-based sensors to exhibit high sensitivity and selectivity toward specific gases. However, pure PTh sensors commonly face challenges such as limited sensitivity, slow response times, and relatively poor stability under varying environmental conditions [94].

To overcome these issues, researchers have developed composite materials by incorporating various nanomaterials with PTh, significantly enhancing its sensing performance. Bai et al. [95] synthesized flexible gas sensors based on ethylenediamine-modified reduced graphene oxide (RGO) combined with PTh through in situ polymerization (Figure 5b). This composite material exhibited approximately four times higher sensitivity to nitrogen dioxide (NO_2_) gas compared to pristine PTh sensors, achieving a detection limit as low as 0.52 ppm. Additionally, the flexible nature of the sensor facilitated its integration into wearable devices, demonstrating its potential for practical environmental monitoring applications. Belhousse et al. [18] further illustrated the importance of optimizing PTh layer thickness in sensor design by fabricating sensors using electrochemically polymerized PTh layers on porous silicon (PSi) substrates. They reported that sensors with PTh polymerized over six cycles provided optimal sensitivity, rapid response, and superior stability when detecting carbon dioxide (CO_2_) and cigarette smoke at room temperature. This highlights the critical role of precise structural control in improving the sensing performance of PTh-based sensors.

**Figure 5 sensors-25-02724-f005:**
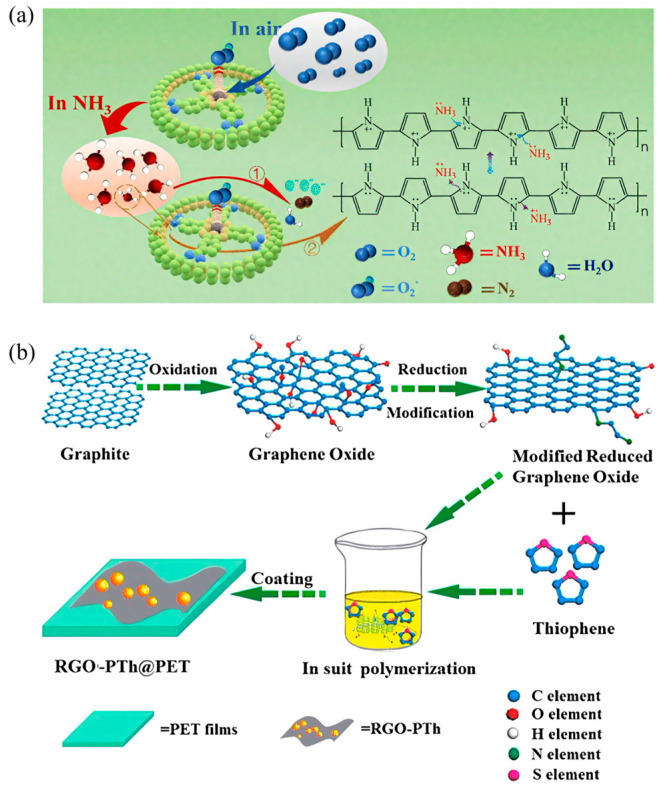
(**a**) Response mechanism of the PPy-TcCoPc sensor to NH_3_ [84]; (**b**) schematic diagram of the preparation process for RGO-PTh hybrid [95].

Future research on PTh-based gas sensors should continue exploring advanced composite formulations and sophisticated fabrication techniques to further enhance sensitivity, selectivity, and stability. Developing flexible, wearable, and cost-effective PTh sensors capable of reliable operation under diverse environmental conditions will significantly broaden their applications in industrial safety, environmental monitoring, and biomedical fields.

#### 2.2.4. PEDOT

Poly(3,4-ethylenedioxythiophene) (PEDOT) is a widely recognized CP known for its excellent electrical conductivity, environmental stability, biocompatibility, and ease of processability, making it highly attractive for gas sensing applications [96]. PEDOT’s conjugated molecular structure provides efficient electron transport pathways, enabling rapid electrical signal generation upon interaction with analyte gases [97]. Furthermore, its good chemical stability and resistance to oxidation make PEDOT suitable for sensors operating under various environmental conditions [98]. However, pristine PEDOT-based gas sensors face certain limitations, including susceptibility to humidity-induced interference, moderate sensitivity, and potential baseline drift.

To overcome these issues, researchers have employed composite approaches and structural optimizations. Xiao et al. [99] successfully developed ammonia sensors by creating core-shell structured nanofibers (Figure 6a), incorporating PEDOT as the core sensing material and polyvinylidene fluoride-trifluoroethylene (PVDF-TrFE) as the protective hydrophobic shell. This design effectively reduced humidity interference while maintaining excellent sensitivity to ammonia. Additionally, PEDOT has been effectively combined with nanostructured materials, such as graphene oxide (GO) and metal nanoparticles, to further enhance sensor sensitivity and selectivity. These composite structures significantly improve sensor performance by increasing active surface area, facilitating gas adsorption, and enhancing electron transport properties [100].

Future research directions for PEDOT-based gas sensors include further composite engineering to boost sensitivity and selectivity, developing robust structures to minimize environmental interference, and exploring novel doping strategies to optimize electronic properties [101]. Additionally, fabricating flexible and wearable PEDOT sensors is expected to expand their applicability significantly across environmental monitoring, industrial safety, and medical diagnostics.

#### 2.2.5. PEDOT:PSS

Poly(3,4-ethylenedioxythiophene):poly(styrene sulfonate) (PEDOT:PSS) is a prominent CP widely applied in gas sensors due to its exceptional electrical conductivity, optical transparency, excellent environmental stability, and solution processability [20]. Incorporating the hydrophilic polyelectrolyte poly(styrene sulfonate) (PSS) significantly enhances the water solubility and processability of PEDOT, facilitating fabrication into various sensor architectures through simple methods like spin-coating, spray-coating, or printing techniques [102].

Despite these advantages, pristine PEDOT:PSS sensors still encounter certain limitations, such as moderate sensitivity and susceptibility to environmental humidity interference, which can affect sensor accuracy and stability. Therefore, various strategies have been explored to improve the sensing capabilities of PEDOT:PSS, including compositing with other nanomaterials and employing structural optimizations. For instance, Alves et al. [103] successfully enhanced PEDOT:PSS gas sensor performance by integrating graphene oxide (GO) into the polymer matrix. This PEDOT:PSS/GO composite sensor exhibited significantly improved sensitivity toward methanol, displaying approximately 2.5 times higher response compared to the pure PEDOT:PSS sensor. Additionally, it maintained stable detection performance across a broad humidity range (0–80% RH) and temperatures (21–60 °C), demonstrating practical applicability in complex indoor environments. In another innovative approach, Farea et al. [104] developed PEDOT:PSS/poly(p-methoxyaniline) (PEDOT:PSS/PPA) nanocomposite sensors for carbon monoxide (CO) detection. The composite structure enhanced the selective interaction with CO molecules, significantly improving the sensor’s selectivity and sensitivity. The sensor displayed remarkable repeatability, stability, and selectivity, effectively discriminating CO from interfering gases such as acetone and toluene (Figure 6b). However, sensor performance was influenced by humidity, indicating the necessity for additional structural modifications or protective layers to mitigate environmental interferences (Figure 6c).

Future research on PEDOT:PSS-based gas sensors should focus on further enhancing sensor selectivity and stability, particularly by engineering composite structures or developing advanced encapsulation strategies to minimize environmental interference [103]. Expanding application scenarios, such as wearable or flexible sensing devices, is another promising direction that leverages the polymer’s intrinsic flexibility and robust mechanical properties, broadening its practical applications in environmental monitoring, healthcare, and industrial safety.

**Figure 6 sensors-25-02724-f006:**
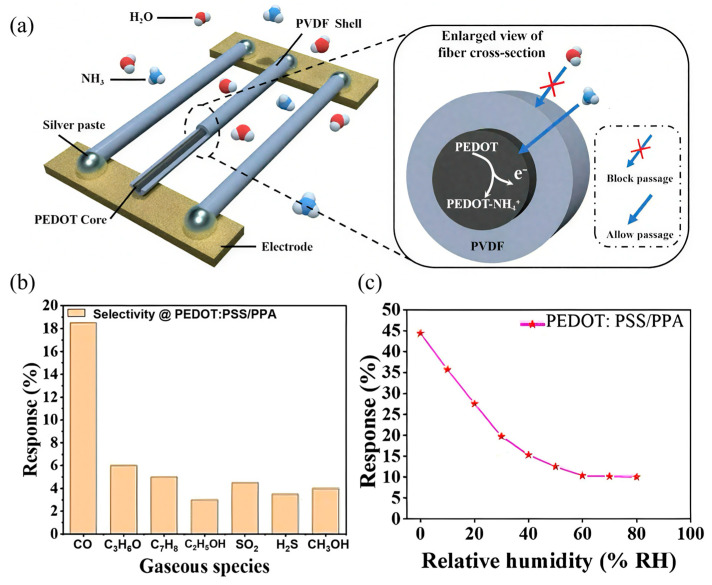
(**a**) Schematic diagram of the CSNF sensor structure and its response principle [99]; sensing performance of the PEDOT:PSS/PPA sensor at 100 ppm of CO. (**b**) Selectivity, (**c**) humidity effect on the PEDOT:PSS/PPA sensor [104].

Overall, conductive polymers such as PANi, PPy, PTh, PEDOT and PEDOT:PSS exhibit multiple performances in gas sensing. The sensing mechanisms, target gases, and response times of different materials are significantly different, as shown in Table 1.

**Table 1 sensors-25-02724-t001:** Comparison of the critical application parameters of different CPs-based sensors for special gas analytes.

Mechanism	Sensor Materials	Target	LOD	Responce	Ref.
Chemiresistive	PEDOT:PSS/PPA	CO	50 ppm	58 s	[104]
Chemiresistive	PANI/SnO_2_	C_6_H_6_	0.4 ppm	33 s	[105]
Mechanical	poly(DTCPA-*co*-BHTBT)-CB	C_7_H_8_	4 ppm	36.47 s	[48]
Chemiresistive	TSP-nAu-PANi	CHCl_3_	–	360 s	[106]
Chemiresistive	GO:PEDOT:PSS	CH_3_OH	–	24.47 s	[103]
Optical	POF/PPy	C_2_H_5_OH	140 ppm	5 s	[61]
Chemiresistive	SWCNT/C_4_F-PPy	C_3_H_6_O	1 ppm	750 s	[107]
Chemiresistive	SnO_2_/PTh	C_3_H_6_O	0.5 ppm	10 s	[35]
Chemiresistive	PANi/AgNWs/Silk	TMA	1.38 ppm	90 s	[24]
Chemiresistive	PANi	H_2_	1 ppm	15 s/17 s	[78]
Chemiresistive	CA-PANi	H_2_S	1 ppm	1 s	[16]
Chemiresistive	PANi/MO_X_	H_2_S	<100 ppm	10 s	[88]
FET	Si/PANI:DBSA	NH_3_	–	2 s	[66]
Chemiresistive	PEDOT-PVDF	NH_3_	10 ppm	80 s	[99]
Chemiresistive	PA-PANI/GO	NH_3_	25 ppm	–	[73]
Chemiresistive	f-MWCNT-PEDOT:PSS	NH_3_	<10 ppm	228 s	[108]
Chemiresistive	PPy-TcCoPc	NH_3_	50 ppm	8 s	[84]
Chemiresistive	CdS QDs-PTh	NH_3_	10 ppm	0.6 s	[36]
Electrochemical	PEDOT:PSS/IrOx Ps	NH_3_	8 ppm	87 ± 9 s	[30]
Chemiresistive	PANi	NH_3_	2.5 ppm	110 s	[22]
Chemiresistive	PPy@LIG	NH_3_	1 ppm	450 s	[86]
Chemiresistive	PANi/FMWCNT	NH_3_	1 ppm	15 s	[77]
Chemiresistive	PANi-MWCNTs/PDMS	NH_3_	10 ppb	–	[109]
Chemiresistive	RGO-PTh	NO_2_	0.52 ppm	498 s	[95]
Chemiresistive	MWCNTs/PANi	NH_3_	0.3 ppm	21 s	[76]
Piezoelectric	GO:PEDOT:PSS	NO_2_	175 ppb	35 s	[43]
Chemiresistive	PTh	NO_2_	0.25 ppm	4980 s	[110]
Chemiresistive	PANi/BP	NO_2_	<2 ppm	98 s	[111]
Chemiresistive	Au-ZnO-PANi	NO_2_	<10 ppm	600 s	[112]

### 2.3. Main Constituent Materials of the Sensor Device

#### 2.3.1. Substrate

Substrates play a crucial role in CPs-based gas sensors, providing essential physical support and significantly influencing sensor performance, sensitivity, stability, and applicability. Ideal substrates should exhibit compatibility with sensing materials, mechanical flexibility, chemical inertness, appropriate thermal stability, and cost-effectiveness [113]. Common substrates include ceramics, polymers, and flexible materials.

Ceramic substrates are widely utilized due to their excellent chemical stability, high-temperature resistance, and mechanical robustness, making them suitable for harsh environmental conditions or sensors requiring high-temperature processing [114]. Feng et al. [105] employed ceramic substrates in PANi/SnO_2_ hybrid gas sensors, taking advantage of ceramics’ high thermal stability and chemical inertness, effectively ensuring stable performance in detecting ammonia and benzene vapor. However, ceramic substrates possess inherent brittleness, limiting their application in flexible electronics and wearable sensors.

Thus, polymer-based substrates, particularly polyethylene terephthalate (PET), have emerged as promising alternatives. PET substrates offer distinct advantages, including excellent flexibility, durability, and compatibility with diverse processing techniques such as printing and coating methods [115]. Boonthum et al. [108] demonstrated an ammonia gas sensor fabricated from functionalized multi-walled carbon nanotubes (f-MWCNTs) combined with PEDOT:PSS on a flexible PET substrate. This sensor exhibited stable performance even when subjected to bending with various curvature radii (Figure 7a), demonstrating significant potential for flexible and wearable sensor applications. Another emerging substrate material is polyimide (PI), characterized by high thermal stability, superior mechanical properties, and excellent chemical resistance, particularly suitable for sensors involving high-temperature processes or requiring enhanced mechanical strength [116]. Kang et al. [107] fabricated flexible sensors using PI substrates interdigitated with Cu/Ni/Au electrodes. This design ensured strong adhesion between electrodes and substrates, enhancing sensor reliability and overall durability.

Future research in substrate materials should focus on developing multifunctional and composite substrates that further enhance sensor performance, flexibility, and environmental resilience. Advancing substrate technologies will significantly expand the practical applications of CPs-based sensors in environmental monitoring, healthcare, wearable electronics, and industrial safety.

**Figure 7 sensors-25-02724-f007:**
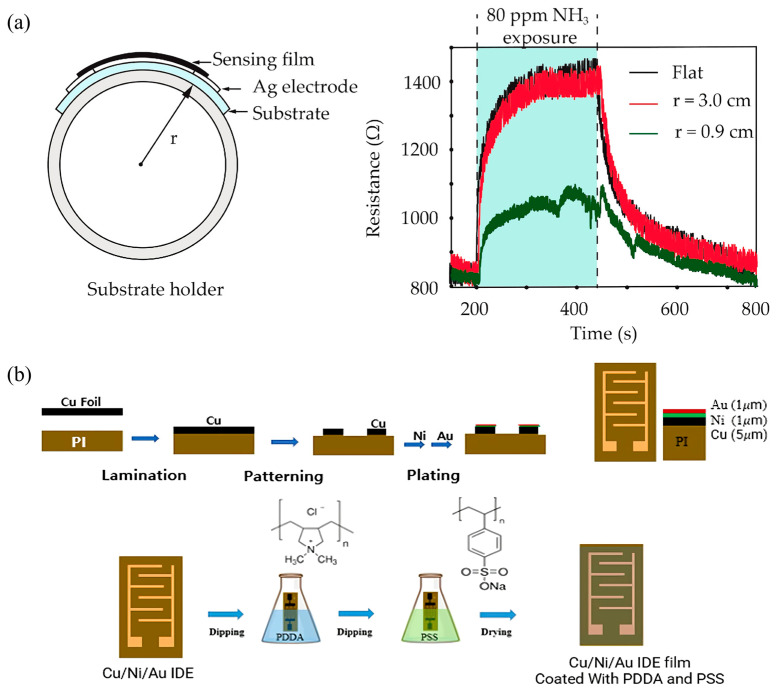
(**a**) Schematic diagram of f-MWCNT-PEDOT:PSS gas sensor under bending test and resistance changes in gas sensor measured during flat and bending (r = 0.9 cm, r = 3.0) states [108]; (**b**) schematic showing the preparation of the PI substrates interdigitated with the Cu/Ni/Au electrodes (IDE substrate) [107].

#### 2.3.2. Electrode Material

Electrode materials play a critical role in CPs-based gas sensors, directly impacting their sensitivity, selectivity, stability, and overall performance [117]. Ideal electrode materials should possess high electrical conductivity, excellent chemical stability, strong adhesion to the substrate, compatibility with CPs, and resistance to environmental interferences.

Noble metals, such as gold (Au), platinum (Pt), and silver (Ag), are extensively used electrode materials due to their exceptional electrical conductivity, chemical inertness, and stability under various environmental conditions [118]. Park et al. [110] developed gas sensors utilizing platinum (Pt) interdigitated electrodes combined with polythiophene (PTh) for nitrogen dioxide (NO_2_) detection. The Pt electrodes provided stable and efficient electron transport pathways, ensuring accurate signal acquisition with minimal interference. However, noble metals’ high cost limits their large-scale practical application, prompting researchers to seek cost-effective alternatives.

Carbon-based materials, including graphene, carbon nanotubes (CNTs), and laser-induced graphene (LIG), have emerged as promising electrode materials due to their outstanding electrical conductivity, large specific surface area, and excellent mechanical flexibility [119]. Kang et al. [107] designed flexible acetone sensors using single-walled carbon nanotube (SWCNT)-based composite materials with Cu/Ni/Au tri-layer electrodes on polyimide (PI) substrates (Figure 7b). The copper (Cu) provided low-cost conductive paths, nickel (Ni) effectively prevented oxidation and migration, while gold (Au) ensured superior electrical conductivity and chemical stability. This multi-layer electrode configuration significantly enhanced sensor performance and stability.

Transparent conductive oxide electrodes, such as indium tin oxide (ITO), have also gained attention, particularly in optical and transparent sensor applications, owing to their combination of transparency and electrical conductivity. ITO electrodes enable real-time optical and electrical signal monitoring simultaneously, broadening their practical applications in multifunctional sensors [120].

Future research should focus on developing composite electrodes and advanced electrode architectures that enhance sensor sensitivity, stability, and durability [121]. Additionally, efforts should be directed toward identifying alternative cost-effective materials and fabrication methods to facilitate large-scale production and commercialization of CPs-based gas sensors.

#### 2.3.3. Packaging Material

Packaging materials are crucial components of CPs-based gas sensors, providing essential protection and isolation from environmental factors, thereby enhancing sensor stability, reliability, and operational lifespan [122]. Suitable packaging materials should exhibit excellent chemical and thermal stability, mechanical strength, effective moisture resistance, and compatibility with CPs to maintain sensor sensitivity and response consistency [123]. Commonly used packaging materials include epoxy resins, silicone rubbers, polyimide (PI), and biodegradable polymers like polycaprolactone (PCL) [124]. Epoxy resins are widely favored for their strong adhesion, high mechanical robustness, and chemical inertness, effectively protecting sensors from harsh environmental conditions. Silicone rubbers offer superior elasticity, excellent resistance to temperature variations, and chemical stability, making them suitable for flexible sensor applications.

For instance, Safaee et al. [125] utilized polycaprolactone (PCL) as a packaging material combined with single-walled carbon nanotubes (SWCNTs) to fabricate wearable optical microfiber textile sensors for real-time monitoring of hydrogen peroxide in biomedical applications (Figure 8a). PCL was selected for its excellent biocompatibility, chemical stability, and ease of fabrication, enabling secure encapsulation and reliable long-term performance in biological environments [126]. Nevertheless, PCL’s relatively slow degradation rate and limited mechanical robustness under high stress conditions present challenges that must be considered in specific applications [127]. Polydimethylsiloxane (PDMS) is another prominent packaging material, particularly beneficial in flexible and wearable sensors due to its outstanding flexibility, biocompatibility, and environmental stability [128]. Lu et al. [129] developed flexible strain sensors using silver nanowires (AgNW), thermoplastic polyurethane (TPU), and PDMS (Figure 8b). PDMS provided critical mechanical flexibility, effectively protecting the sensor under repeated mechanical deformation while maintaining stable sensing performance.

Future research on packaging materials should focus on developing multifunctional composites and advanced packaging techniques, such as encapsulation strategies to minimize environmental interference and enhance durability [130]. Efforts to improve biocompatibility, biodegradability, and mechanical resilience will significantly broaden the practical applications of CPs-based sensors, particularly in wearable devices, healthcare, and environmental monitoring sectors.

**Figure 8 sensors-25-02724-f008:**
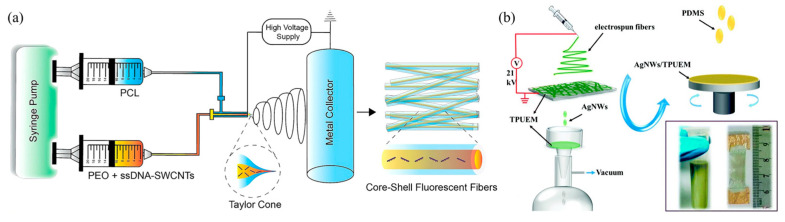
(**a**) Core-shell electrospinning setup for the fabrication of the optical microfibrous textiles [125]; (**b**) schematic diagram of ATP strain sensor production [129].

## 3. Application of CPs-Based Gas Sensors

### 3.1. Promising and Broad Applications

#### 3.1.1. Environmental Monitoring Field

With increasing environmental pollution challenges, accurate and efficient environmental monitoring has become crucial for safeguarding human health and ecological sustainability. CPs-based gas sensors have emerged as powerful tools in environmental monitoring due to their unique properties, including high sensitivity, rapid response, ease of fabrication, and compatibility with flexible and wearable technologies. These characteristics position CPs as ideal candidates for monitoring harmful gases such as nitrogen dioxide (NO_2_), sulfur dioxide (SO_2_), ammonia (NH_3_), volatile organic compounds (VOCs), and greenhouse gases [1,2,3].

PEDOT:PSS, a widely studied CP, has shown remarkable effectiveness in detecting environmentally harmful gases due to its superior electrical conductivity and chemical stability [131]. For example, Deller et al. [21] developed a voltammetric sensor by integrating PEDOT:PSS with gold nanoparticles (AuNPs) to detect pirimicarb (PMC), an environmental contaminant. This sensor demonstrated high sensitivity and selectivity, with a low detection limit and stable performance, highlighting the applicability of CPs in precise environmental pollutant monitoring.

Polyaniline (PANi) and its composites also exhibit great potential in environmental gas monitoring. Tang et al. [111] designed a room-temperature NO_2_ sensor by combining PANi with black phosphorus (BP), achieving excellent sensitivity within a concentration range of 2–60 ppm NO_2_. Similarly, Bonyani et al. [112] fabricated a highly selective NO_2_ sensor based on Au-modified ZnO-PANi composite nanofibers (Figure 9a), significantly improving sensor performance and reliability.

Despite these advancements, CPs-based environmental sensors still face challenges regarding stability under variable humidity and temperature conditions, long-term operational durability, and cross-sensitivity to interfering gases. Future research directions include developing advanced composite materials, optimizing sensor architectures, and incorporating intelligent data-processing methods to further enhance sensor sensitivity, selectivity, and environmental robustness [132]. Additionally, efforts toward flexible, miniaturized, and wearable sensor systems will enable widespread deployment for continuous, real-time environmental monitoring, substantially improving air-quality management and pollution control strategies.

#### 3.1.2. Industrial Production Field

In industrial production, accurate and real-time monitoring of gas concentrations is essential to ensure product quality, enhance operational safety, and prevent potential accidents caused by hazardous gases. CPs-based gas sensors have gained increasing attention in industrial applications due to their advantages, such as rapid response, high sensitivity, low fabrication cost, and the ability to function effectively at room temperature. Volatile organic compounds (VOCs), including benzene, toluene, and formaldehyde, are common pollutants generated in various industrial processes, posing severe health risks and environmental hazards [133].

CPs gas sensors, owing to their excellent selectivity and sensitivity, have shown significant promise for VOC monitoring in manufacturing environments. For instance, Selvanayakam et al. [106] developed a sensor based on tamarind seed polysaccharide (TSP)-coated gold nanoparticles integrated with polyaniline (PANi) for detecting chloroform vapor in industrial emissions. This composite sensor displayed exceptional sensitivity, rapid response, and robust selectivity, effectively distinguishing chloroform from other interfering gases. Moreover, CPs-based gas sensors are widely employed in detecting flammable gases such as hydrogen, essential for safety monitoring in chemical industries and energy sectors [109]. Dipak et al. [22] demonstrated a hydrogen sensor using PANi nano-ink with a response of about 75% to ammonia (Figure 9b), which can effectively detect very small amounts of ammonia in the environment, providing crucial safety assurance in hydrogen-based energy applications.

Despite these advantages, challenges remain, including the susceptibility of CPs sensors to humidity interference, limited long-term stability under harsh industrial conditions, and potential baseline drift over extended use. Addressing these issues through material engineering, improved encapsulation techniques, and sensor design optimization is essential for advancing practical applications.

Future research should focus on developing advanced composite materials, optimizing sensor design, and integrating intelligent data-processing methods to enhance sensor reliability, stability, and resilience to environmental interference. Additionally, exploring flexible, miniaturized, and low-cost CPs sensors suitable for integration into industrial automation systems will significantly expand their applicability and impact on industrial safety and quality control.

#### 3.1.3. Food Safety Field

Gas sensing plays a crucial role in ensuring food safety, acting as an “invisible guardian” throughout the entire food supply chain—from production to sale—to continuously safeguard food quality. When food begins to spoil, various volatile gases are released. With its high sensitivity, gas-phase sensing technology can rapidly detect these gases, effectively assessing the freshness and safety of food products.

As the most widely consumed meat globally, pork serves as an essential source of animal protein for humans. However, fresh pork is highly susceptible to spoilage during storage. Microorganisms and enzymes can accelerate the decomposition of proteins, fats, and other components, producing biogenic amines such as ammonia, trimethylamine (TMA), and dimethylamine [134,135,136]. The deterioration of meat quality typically results in noticeable changes in surface texture, color, and odor. Among the numerous indicators used to evaluate pork freshness, TMA is particularly important. Its concentration gradually increases as pork freshness declines, and it emits a pungent rancid odor closely associated with the degree of spoilage. Consequently, accurate TMA detection can effectively and intuitively reflect pork freshness, making it a critical parameter in pork quality assessment.

To address this need, Li et al. [24] synthesized polyaniline (PANi) and silver nanowires (AgNWs) onto silk fibroin fibers (SFF) through an in-situ polymerization method, fabricating a novel, reliable, flexible, and easy-to-use gas sensor. This sensor was employed to detect various gases at 100 μg/L, including TMA, NH_3_, H_2_S, H_2_O, and C_2_H_6_O (Figure 9c). Compared to other gases, the sensor exhibited a significantly higher response toward TMA, allowing rapid, non-destructive, sensitive, and cost-effective detection of TMA in pork. Thus, it effectively evaluates pork freshness, preventing consumers from ingesting spoiled meat and ensuring food safety.

Compared with traditional detection methods, gas sensors offer advantages such as simpler operation and shorter detection times. They can provide real-time monitoring of meat freshness throughout food production, processing, transportation, and sales, enabling the timely identification of spoiled products and minimizing economic losses. Consequently, these sensors represent efficient tools for freshness detection in the food industry and contribute significantly to the advancement of food safety technologies.

#### 3.1.4. Medical Diagnostic Field

Accurate, non-invasive, and rapid medical diagnostics significantly enhance the quality of patient care and facilitate early disease detection and monitoring [137]. CPs-based gas sensors offer substantial potential for medical diagnostics due to their high sensitivity, rapid response times, portability, flexibility, and ability to operate at room temperature, making them ideal for analyzing biomarkers in exhaled breath or volatile organic compounds (VOCs) emitted from the human body.

Exhaled breath analysis has become an essential diagnostic approach, providing vital insights into metabolic disorders, infectious diseases, and various physiological conditions. For instance, ammonia in human breath is an important biomarker for the diagnosis of kidney disease. CPs sensors have demonstrated remarkable efficacy in accurately and rapidly detecting trace ammonia levels due to their adjustable conductivity and selectivity [138,139]. Zhu et al. [133] developed a flexible ammonia sensor that uses a combination of polyaniline (PANi) and multi-walled carbon nanotubes (MWCNTs) to achieve excellent sensitivity and accurately detect ammonia in exhaled breath, providing a reliable technical means for non-invasive diagnosis of kidney disease. Moreover, sensors based on CPs have been successfully applied to detect acetone levels in the breath, an important marker of diabetes. Ananda et al. [140] prepared a gas-phase sensor based on PPy composites with ternary oxide ZnCo_2_O_4_ (ZCO) and MnCo_2_O_4_ (MCO) nanoparticles. Both nanocomposites exhibit high sensitivity and selectivity to acetone at room temperature (Figure 9d). The sensor effectively distinguished diabetic breath samples from healthy controls, indicating its considerable potential in non-invasive diabetes monitoring.

Despite these advancements, CPs-based gas sensors still face challenges such as cross-sensitivity to environmental factors, limited long-term stability, and potential signal drift. Future research should focus on enhancing sensor specificity through molecular engineering, integrating advanced composite materials, developing robust sensor designs, and incorporating artificial intelligence for accurate, real-time diagnostics. Further efforts toward creating wearable, low-cost, and flexible diagnostic devices will significantly expand the scope and impact of CPs-based sensors in medical applications, driving their adoption in personalized medicine and continuous health monitoring.

Conductive polymer-based gas sensors are widely used in multiple fields. Materials such as PANi, PPy, PTh, PEDOT, and PEDOT:PSS have different properties. Table 1 summarizes the sensing mechanisms, target gases, and performance data of different materials. Due to diverse application requirements involving multiple dimensions such as sensitivity and stability, and the performance of materials being affected by processes, it is impossible to simply compare which material is the best.

**Figure 9 sensors-25-02724-f009:**
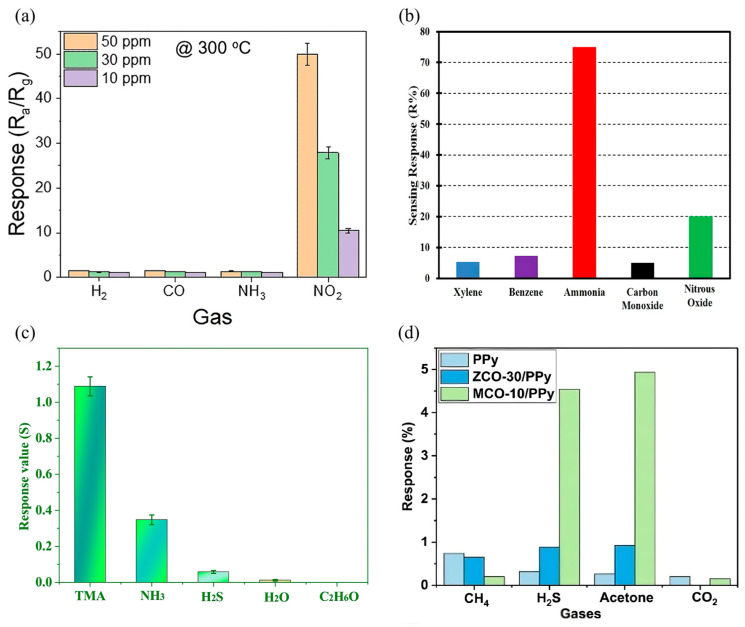
(**a**) Selectivity graph of Au-decorated ZnO-PANi (25 wt.%) composite nanofiber gas sensor to interfering gases at 300 °C [112]; (**b**) selectivity of the PANi nano gas sensors [22]; (**c**) sensor response of the PANi/AgNWs/silk composite nanofibers to 100 μg/L of different gases [24]; (**d**) response of PPy, ZCO-30/PPy, and MCO-10/PPy for various gasses along with a comparative response [140].

### 3.2. Critical Issues in Real Applications

#### 3.2.1. Long-Term Stability and Humidity Drift: Mechanisms and Quantitative Evaluations

Although CPs-based gas sensors can be used to detect multiple gases and exhibit excellent performances, in practical application scenarios, they still face two key challenges: humidity-induced drift and poor long-term stability. Especially when operating in a variable environment for a long time, these problems seriously restrict their reliable working. The root cause of the performance degradation lies in the chemical, electrochemical, and mechanical changes that occur at the interface between the CPs-based sensing layer and the matrix electrode.

In a humid environment, due to the possible cross-reaction between different gases, such sensors may produce similar responsiveness to non-target gases. Therefore, the gas sensing performance may be a mixed response to the target gas and water vapor [141]. On the one hand, water adsorption consumes the adsorption sites of sensing materials, leading to the misjudgment of the concentration of target gases and reducing measurement accuracy. On the other hand, water molecules could interact with polymer chains and disrupt the charge transport path. In addition, the oxidizing or reducing gases may react with the main molecular chain of CPs or their composites, to destroy the conjugated system, reduce the carrier density, and weaken the conductivity and sensing ability of the sensor [142]. For example, in PEDOT:PSS, the PSS shell layer will adsorb water molecules, causing volume expansion, leading to an increase in the distance between PEDOT cores, and thus reducing the carrier transmission efficiency, and further hindering the target gas from reaching the sensing site, prolonging the response time and reducing the sensitivity [143]. By designing a multi-dimensional structure or compounding with inorganic materials, the adsorption sites and charge transfer efficiency of CPs-based sensing materials can be effectively enhanced, thereby improving the responsiveness and selectivity of related sensors. For example, inorganic two-dimensional (2D) nanomaterials such as rGO and MXenes have been introduced to improve their electrical conductivity and enhance their mechanical properties [73,95], using other polymers with good hydrophobicity and thermal stability, such as PDMS and fluorinated polyurethane (PU) to encapsulate these sensors [121,124,125].

Another important challenge for gas sensors is their long-term stability. This not only exists in CPs-based but also in other sensors. This is because during long-term use, sensing active materials may experience performance degradation or even failure due to mechanical external forces, environmental erosion, aging, and other factors. For example, when CPs-based sensors are exposed to air for a long time, due to the obvious dedoping effect, the adsorption sites of CPs may be affected, resulting in a significant decline in sensing performance. At the same time, the presence of oxygen could cause the degradation of CPs, thereby reducing their conductivity [144]. In flexible or wearable devices, mechanical stress cycling may cause delamination at the interface between the CPs films and the substrates, leading to interface failure and functional loss, finally affecting the related sensing performances and shortening the service life of such sensors.

#### 3.2.2. Biocompatibility, Toxicology, and Regulatory Considerations for CP-Based Wearable and Biomedical Sensors

CPs, especially PANi, PPy, PThs, PEDOT, and PEDOT:PSS, have been widely explored for wearable and biomedical sensing platforms. However, translating these materials from laboratory prototypes to clinically viable devices necessitates careful consideration of biocompatibility, cytotoxicity, and regulatory compliance under prolonged skin contact or implantation scenarios, as shown in Table 2.

##### Biocompatibility and Toxicological Assessment

PANi, while conductive and easy to process, may release toxic degradation products (e.g., aniline) if not adequately stabilized. Doping agents and processing residues significantly influence its biocompatibility, Composites or coatings (e.g., with biopolymers like chitosan or silk fibroin) are often employed to improve its safety [145,146].

PPy has demonstrated favorable in vitro and in vivo biocompatibility, so it is generally considered non-cytotoxic, especially when synthesized electrochemically without residual monomers or toxic dopants. Several studies show that PPy-coated scaffolds support cell adhesion and proliferation [147,148].

PThs also have excellent biocompatibility, with few adverse reactions in cell culture and in vivo models. Therefore, they can also promote the adhesion and differentiation of neural stem cells and have significant potential in neural regeneration and bioelectronic devices that require long-term tissue integration [149,150].

PEDOT and PEDOT:PSS, especially when purified or treated to remove excess PSS (e.g., with DMSO or ethylene glycol), exhibit low cytotoxicity and good compatibility with fibroblasts and neuronal cells [151]. As a result, PEDOT:PSS has been used in bioelectronic implants, such as cochlear [152] and neural interfaces [153], supporting its suitability for extended contact with biological tissues.

##### Relevant Standards and Regulatory Framework

ISO 10993 Series [154]: For any device involving skin contact (>30 days) or implantation, ISO 10993 mandates a series of biological evaluations including:(1)ISO 10993-5 (in vitro cytotoxicity);(2)ISO 10993-10 (skin irritation and sensitization);(3)ISO 10993-11 (systemic toxicity);(4)ISO 10993-6 (implantation effects) if the device is implanted.

FDA Device Classification [155]:

Wearable CPs-based sensors typically fall under Class I or Class II medical devices, depending on the intended use (diagnostic vs. therapeutic) and level of invasiveness. Implantable CPs-based sensors, if developed, would likely be Class III (requiring premarket approval), especially for glucose, neurotransmitter, or gas biomarker detection. 

Material Risk Assessment: 

For example, PEDOT:PSS was evaluated by the U.S. FDA as part of neural recording systems and passed biocompatibility testing in several device submissions (e.g., for cortical electrodes).

##### Design Considerations for Clinical Translation

For clinical translation, there are many serious considerations for the material systems design on CPs-based gas sensors, as shown below.

(1)Encapsulation strategies using biocompatible elastomers (e.g., PDMS, TPU) are essential to isolate CPs from direct tissue exposure while maintaining sensing functionality [125];(2)Incorporation of bioinert and hydrophobic coatings also helps prevent ion leaching and immune response [156];(3)Long-term implantation trials (animal models >30 days) are necessary to assess chronic inflammation and fibrotic encapsulation [157].

**Table 2 sensors-25-02724-t002:** Some biosafety and regulatory standards on CPs.

Conducting Polymer	Biocompatibility	Toxicity Concerns	Regulatory Standard	Ref.
PANi	Variable; can release toxic aniline, modified forms, or composites safer	Aniline toxicity, residual dopants, and degradation products are concerns	ISO 10993-5, -10; FDA Class I/11 depending on application	[145,146]
PPy	Generally good; supports cell adhesion and proliferation	Low toxicity if properly synthesized	ISO 10993-5, -10, -11; preclinical animal studies needed for implants	[147,148]
PTh	General, modification or compounding for improvement	The toxicity of degradation products is unknown; residual monomers may be toxic	ISO 10993-5, -10; in some cases, additional in vitro and in vivo testing may be required based on application	[149,150]
PEDOT	Excellent; used in neural and cardiac interfaces	Minimal; depends on dopants and processing additives	ISO 10993-1, -5, -6, -10, -11; used in FDA-cleared implants	[151]
PEDOT:PSS	Good after PSS removal or treatment (DMSO/EG); low cytotoxicity	Excess PSS may irritate; removal improves safety	ISO 10993-5, -10; reviewed under FDA Class 11 submissions (e.g., neural devices)	[152,153]

## 4. Conclusions and Prospects

CPs-based gas sensors hold significant promise across numerous fields due to their unique combination of tunable electrical/chemical properties and organic nature, providing versatile solutions for gas detection. This review has presented a comprehensive understanding of this class of sensors by examining their sensing mechanisms, sensitive materials, and device components. CPs gas sensors leverage diverse transduction mechanisms—including electrochemical, chemiresistive, optical, piezoelectric, and field-effect transistor (FET) methods—each offering distinct advantages suited to specific applications. Likewise, a variety of CP materials (such as PANi, PPy, PTh, PEDOT:PSS, and related composites) have been explored, each exhibiting unique gas-responsive characteristics and selectivity toward certain analytes. Key device components (e.g., flexible substrates, electrode configurations, and encapsulation materials) also critically influence sensor performance and durability. By integrating insights from detection principles, material properties, and device design, researchers have greatly advanced CPs-based gas-sensing technology and expanded its practical applicability.

Despite these advances, several challenges and opportunities remain, which define important directions for future research and development. First, improving sensor sensitivity and selectivity continues to be paramount. In complex gas environments, CPs sensors can be vulnerable to interference from other gases, which complicates the accurate identification of target species. Future studies should delve deeper into the interaction mechanisms between CPs materials and gas molecules and optimize polymer structures (e.g., nanoscale morphology and doping) to enhance selective adsorption and reaction with specific gases. Strategies such as incorporating nanostructured additives or forming CPs nanocomposites can create additional reactive sites and more efficient charge-transfer pathways, thereby dramatically boosting sensor response. For instance, integrating CPs with high-surface-area nanomaterials has been shown to improve the detection of gases like H_2_S by increasing adsorption sites and electron transport efficiency.

Second, long-term stability and reproducibility are critical issues that need to be addressed. Variations in environmental conditions (temperature, humidity, etc.) often affect CPs sensor baselines and response, leading to drift and inconsistent results over time. To overcome this, it is essential to develop robust sensor designs that minimize environmental susceptibility. For example, through innovative packaging materials and coatings that shield the sensitive layer from ambient fluctuations. Refining fabrication processes to produce uniform and stable polymer films, as well as implementing calibration or compensation techniques, will help ensure consistent performance. Improving the environmental adaptability of CPs sensors (for instance, by integrating humidity/temperature compensation elements) can greatly enhance their reliability in real-world applications, thus improving repeatability and facilitating widespread deployment.

Third, the emergence of intelligent gas-sensing systems is a frontier area poised to elevate CPs-based sensor capabilities. By coupling sensor arrays with advanced algorithms (such as machine learning and artificial intelligence), researchers can create electronic nose systems capable of recognizing complex odor/gas patterns and distinguishing specific target gases within mixtures. Machine learning techniques can analyze the multidimensional data from CPs sensor arrays, filter out interference, and even perform real-time pattern recognition that surpasses the selectivity achievable by materials alone. Recent work has demonstrated that hybrid sensor arrays combined with machine learning can rapidly identify hazardous gases in complex backgrounds, highlighting the power of data-driven approaches for enhancing selectivity and sensitivity. Integrating CPs gas sensors into the Internet of Things (IoT) framework with wireless connectivity and cloud analytics can further enable smart gas monitoring networks that learn and adapt over time. Such intelligent systems represent a promising direction to improve accuracy in complex scenarios (by, for example, self-calibrating for drift or compensating for cross-sensitivity), thereby expanding the practicality of CPs-based sensors in industrial safety, environmental monitoring, and other areas where automated, real-time decision-making is crucial.

Fourth, multi-functional integrated sensing platforms offer another promising avenue for future development. Instead of operating in isolation, CPs-based gas sensors can be combined with other types of sensors (such as temperature, humidity, pressure, or even multiple gas sensors in an array) on a single platform to provide comprehensive environmental data. By monitoring multiple parameters simultaneously, such integrated systems can account for environmental factors and improve overall measurement accuracy and context awareness. For example, real-time readings of ambient temperature and humidity alongside gas concentration allow for automatic compensation of environmental effects on the gas sensor’s output, yielding more reliable results. Moreover, multi-gas sensor arrays can be designed to detect a suite of gases at once, enabling a broad-spectrum “electronic nose” capable of profiling complex gas mixtures (e.g., for air quality or breath analysis). The fusion of data from different sensor modalities can thus enhance the selectivity and robustness of the sensing system. In the future, lab-on-a-chip implementations may integrate CPs gas sensors with microfluidic channels, chemical detectors, and electronic circuitry, culminating in portable devices that offer multi-modal sensing and on-site analysis for applications ranging from environmental surveillance to medical diagnostics.

Fifth, exploring new sensor form factors, particularly flexible, wearable, and implantable gas sensors, is an exciting direction to broaden the applicability of CPs-based sensors. Owing to the intrinsic mechanical flexibility of CPs, there is considerable potential to fabricate gas sensors on bendable substrates (plastics, textiles, or even paper), enabling devices that conform to various surfaces or can be worn on the body. In recent studies, researchers have developed wearable CPs gas sensors (for example, a PEDOT-based ammonia sensor on a textile substrate) that can continuously monitor gaseous biomarkers in human sweat or breath. These flexible and wearable sensors open up opportunities for personalized health monitoring (such as real-time breath analysis for medical screening) and on-body environmental exposure tracking for occupational safety. Looking ahead, implantable CPs gas sensors could be envisioned for specialized biomedical applications. For instance, detecting internal gas biomarkers or changes in blood chemistry in vivo, provided that biocompatible materials and safe operation can be ensured. While implantable gas sensors are still largely conceptual, the combination of CPs’ biocompatibility (in certain formulations) and their compatibility with soft electronics suggests that future research could yield minimally invasive gas-sensing devices for healthcare. Overall, developing flexible, stretchable, and wearable CPs gas sensors will significantly expand their use cases, allowing integration into everyday objects, clothing, or even the human body, thereby extending gas monitoring capabilities to scenarios that were previously impractical for rigid, conventional sensors.

Sixth, there is a growing interest in sustainable and eco-friendly materials for sensor development, and CPs-based gas sensors stand to benefit from this trend. Unlike sensors based on scarce or non-renewable inorganic materials, CPs sensors can potentially be made with renewable or biodegradable components. Future research may focus on designing biodegradable CPs or composites that maintain excellent sensing performance while being environmentally benign after their service life. For example, incorporating natural polymers or biodegradable matrices with CPs blends could produce sensors that eventually decompose under specific conditions, reducing electronic waste. Such sustainable sensors would be especially valuable in disposable or short-term use applications (like wearable health patches or environmental sensors deployed in large numbers) where device recovery is difficult.

Additionally, green synthesis approaches for CPs (using less toxic reagents or energy-efficient processes) and the recycling or reprocessing of CPs materials are important considerations for making the next generation of gas sensors more sustainable. Embracing eco-design principles in developing CPs gas sensors will help align the field with global sustainability and environmental safety goals, without sacrificing the performance improvements gained in recent years. Beyond these technical innovations, interdisciplinary integration of CPs-based gas sensors with other fields will further expand their impact and unlock novel applications. One such field is catalysis, where integrating gas sensors with catalytic processes could greatly benefit real-time reaction monitoring and catalyst development.

In catalytic reactors or chemical synthesis systems, CPs gas sensors can be used to monitor reaction gases in real-time, providing immediate feedback on reaction progress. For example, a CP sensor placed in a reactor could detect the emergence of a particular gaseous product or the depletion of a reactant, enabling dynamic adjustments to reaction conditions (temperature, feed rate, etc.) to optimize yield and selectivity on the fly. Conversely, incorporating catalytic nanoparticles or enzymes into a CP sensor’s sensitive layer can create a hybrid sensor-catalyst material that not only facilitates a specific chemical reaction but simultaneously detects its gaseous products. Such multi-functional composites, possessing both catalytic and sensing capabilities, allow for simultaneous catalysis and monitoring. This dual function is valuable for studying reaction mechanisms; the sensor’s readings give insight into intermediate formation and reaction rates, and it can aid in developing smarter catalysts (since the effectiveness of a catalyst under various conditions can be directly observed through the integrated sensor response).

In summary, the convergence of CPs-based gas sensing with catalysis research offers a powerful approach for advancing process control in chemical manufacturing and for innovating new catalytic materials with built-in sensing functionality. Another domain where CPs gas sensors show great promise is biomedical diagnostics. Gas biomarkers are an emerging frontier in medical screening and monitoring. For instance, volatile organic compounds (VOCs) in exhaled breath can serve as indicators for diseases such as diabetes, lung cancer, or infectious diseases. CPs-based gas sensors, with appropriate selectivity tuning, can be tailored to detect specific biomarker gases, potentially enabling non-invasive diagnostic tools. For example, a CP sensor designed to respond to acetone in breath could assist in monitoring diabetes (as breath acetone levels correlate with blood glucose status), and sensors for nitric oxide or other breath VOCs could help in the early detection of respiratory conditions. By integrating these sensors into portable or wearable formats (such as a handheld breath analyzer or a patch that samples skin-emitted gases), patients could perform real-time health monitoring outside of clinical settings.

Furthermore, coupling CPs gas sensors with microfluidic and MEMS technologies can yield miniaturized lab-on-chip diagnostic platforms capable of analyzing exhaled breath or headspace from biological samples with high sensitivity. Such systems could concentrate trace gases and deliver them to the CP sensor array for detection, providing a fast and accurate analysis for personalized medicine. The interdisciplinary collaboration between material scientists, medical researchers, and engineers will be key to optimizing CPs sensor designs for biocompatibility, selectivity to relevant biomarkers, and user-friendly operation, ultimately bringing gas sensing diagnostics into routine healthcare practice. In the energy sector, the integration of CPs-based gas sensors can significantly enhance the safety and efficiency of energy storage and conversion systems. One promising application is in battery technology: lithium-ion batteries and other high-energy devices can release trace gases (such as CO_2_, CO, or electrolyte vapors) as early indicators of thermal runaway or degradation.

Embedding CPs gas sensors within battery packs or enclosures could allow continuous internal monitoring of such gas evolution, providing early warnings of battery failure or overheating. This real-time sensing could enable proactive measures (like cooling or disconnecting a failing cell) to prevent fires or explosions, thereby greatly improving battery safety and reliability. Similarly, for fuel cells and other fuel-based energy devices, monitoring the concentrations of reactant and product gases (H_2_, O_2_, water vapor, etc.) using CPs sensors can help in optimizing performance. Sensors can feed information to control systems to adjust fuel flow or operating conditions for maximum efficiency and detect deviations that might indicate catalyst poisoning or membrane leaks.

Moreover, incorporating CPs gas sensors in environmental control systems of energy facilities (such as hydrogen storage, biogas plants, or carbon capture units) can aid in leak detection and emissions monitoring, contributing to safer and more sustainable energy operations. These examples in energy applications underscore the broad prospects for CPs gas sensors when combined with energy technology development a synergy that can drive innovations in both fields simultaneously. In summary, CPs-based gas sensors are poised to play an increasingly significant role in future sensing technologies and interdisciplinary applications. By addressing the remaining challenges, improving sensitivity/selectivity, enhancing stability, and embracing intelligent algorithms—and by exploiting new opportunities such as multi-modal integration, flexible form factors, sustainable materials, and cross-domain collaborations, the next generation of CPs-based gas sensors will become even more sensitive, selective, robust, and versatile.

Future researchers should continue to focus on these emerging directions and foster innovation at the intersection of materials science, engineering, and application domains. Through sustained research efforts and interdisciplinary collaboration, CPs-based gas-sensing technology will continue to advance, driving breakthroughs in how we detect and utilize gas information in environmental monitoring, industrial process control, healthcare, energy, and beyond. Ultimately, the ongoing innovations will ensure that CPs gas sensors realize their full potential as key components of smart, responsive, and sustainable sensing systems for the modern world.

In addition, artificial intelligence (AI) technology has great potential in the future development of gas sensors on the basis of CPs. In terms of data processing, AI can efficiently analyze the massive and complex data generated by sensors toward real and unpredictable gas analytes, like E-noses for pneumoconiosis screening and diagnosis [158]. For example, deep learning algorithms could deeply mine the response data of sensors under different gas concentrations and environmental conditions and establish accurate gas recognition models, thereby greatly improving the recognition accuracy of sensors for target gases and effectively reducing misjudgments and omissions. In terms of adaptive adjustment, AI may dynamically adjust the working mode and parameter settings of sensors according to real-time environmental parameters and historical data of sensors. For example, when there are large changes in environmental humidity or temperature, the compensation mechanism of the sensor may be automatically optimized through AI algorithms to keep the sensor in the best detection performance at all times and further improve its stability and reliability in complex and changeable environments. Moreover, with the predictive analysis ability of AI, it is possible to predict in advance possible failures or performance declines of sensors so that timely maintenance and replacement can be carried out to reduce the risk of equipment operation and improve the operating efficiency of the entire gas detection system. By deeply integrating AI technology with CPs-based gas sensors, it is expected to promote leapfrog development in this field and create a more intelligent and efficient new era of gas detection.

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
