# Peer review of "Conducting Polymers-Based Gas Sensors: Principles, Materials, and Applications"

_sensors, 2025, doi:10.3390/s25092724_

Round 1
Reviewer 1 Report
Comments and Suggestions for Authors
This review surveys conducting polymer gas sensors, clearly explaining transduction mechanisms, materials synthesis, and diverse applications. It integrates recent advances and outlines future directions—miniaturization, AI‑enabled analysis, flexible platforms—offering valuable guidance, though critical comparison with competing sensing technologies would strengthen impact. The paper is easy to follow. Some minor concerns require attention in the revision process.
- The review asserts that conducting‑polymer (CP) sensors deliver “extremely high sensitivity” and “excellent selectivity” at low cost compared with metal‑oxide or MOF devices, yet no side‑by‑side quantitative benchmarks (limit of detection, response/recovery time, energy draw) are supplied. Could the authors include comparative tables or a meta‑analysis to substantiate these superiority claims?
- The optical‑sensing subsection cites only a handful of demonstrations and omits many recent advances—e.g., plasmon‑enhanced CP fluorophores, waveguide‑integrated CP interferometers, and cavity‑coupled absorption sensors. Would the authors expand this section with a systematic literature survey (2019‑2025) that tabulates key architectures, figures‑of‑merit, and application domains to give readers a complete state‑of‑the‑art snapshot? ​The following papers may of interest to authors. 10.1364/OE.444323, 10.1021/acsami.3c18702
- Long‑term stability and humidity‑induced drift are acknowledged as persistent obstacles, but the manuscript offers only qualitative comments. What are the dominant chemical or mechanical degradation pathways (e.g., dedoping, backbone oxidation, water sorption) and what quantitative lifetime data exist from accelerated‑aging or field trials?
- Wearable and biomedical applications are highlighted, but the review omits any assessment of biocompatibility, toxicology, or regulatory requirements for prolonged skin contact or implantation of CP materials. What data and standards (ISO 10993, FDA class definitions, etc.) support the safe translation of these sensors to human use? Need more clarification.
Author Response
Response to the reviewer1’s comments
Firstly, we deeply appreciate the reviewer for the hard work on our manuscript. Those comments are all valuable and very helpful for revising and improving our paper. Below you will find our point-by-point response to your comments (Answers are marked in red):
*********************************************************************
Reviewer 1’s comments:
This review surveys conducting polymer gas sensors, clearly explaining transduction mechanisms, materials synthesis, and diverse applications. It integrates recent advances and outlines future directions—miniaturization, AI‑enabled analysis, flexible platforms—offering valuable guidance, though critical comparison with competing sensing technologies would strengthen impact. The paper is easy to follow. Some minor concerns require attention in the revision process.
Response: Thank you very much for your keen and professional identification of our article.
1.The review asserts that conducting‑polymer (CP) sensors deliver “extremely high sensitivity” and “excellent selectivity” at low cost compared with metal‑oxide or MOF devices, yet no side‑by‑side quantitative benchmarks (limit of detection, response/recovery time, energy draw) are supplied. Could the authors include comparative tables or a meta‑analysis to substantiate these superiority claims?
Response: Thank you for this critical comment. This is of great significance for us to further optimize the statement.
It should be pointed that metal oxides, metal-organic frameworks (MOFs), and conductive polymers are all having large and complex material systems, and each type of them shows great performance differences and special application characteristics. Also, due to the different synthesis processes, microstructures, and action mechanisms of each system, their performances are often affected by many factors not only just in preparation conditions and application scenarios. Therefore, it is difficult to conduct a comprehensive and direct comparison of different type of material systems through a single standard or form. In the future, we or other researchers in this field will continue to pay attention to this aspect and deepen our understanding of broad material systems. Maybe, a much longer, more comprehensive and special review about this topic can be find in the future. At this stage, after repeated deliberation and strict check, we have refreshed these inappropriate expressions and provides a new statement in the revised “Introduction” section, as following:
“Metal oxides and metal-organic frameworks (MOFs) have shown excellent adsorption and selective detection performances towards gas sensing, but they often need to work under high temperature conditions [14, 15]. In contrast, the great advantages of CPs in this field are that they can work at room temperature, quickly respond to gases through intermolecular interactions, enable solution processing, have low power consumption, as well as be suitable for flexible integration andportable monitoring [16].”
- The optical‑sensing subsection cites only a handful of demonstrations and omits many recent advances—e.g., plasmon‑enhanced CP fluorophores, waveguide‑integrated CP interferometers, and cavity‑coupled absorption sensors. Would the authors expand this section with a systematic literature survey (2019‑2025) that tabulates key architectures, figures‑of‑merit, and application domains to give readers a complete state‑of‑the‑art snapshot? The following papers may of interest to authors. 10.1364/OE.444323, 10.1021/acsami.3c18702
Response: Thank you for this valuable suggestion. We have systematically reviewed the literature from 2019 to 2025 again, and added relevant literature citations in the revised manuscript, with particular emphasis on supplementing two important papers recommended by you (10.1364/OE.444323, 10.1021/acsami.3c18702).
In the section of “2.1.5. Optical sensing” of the revises manuscript:
“In recent years, in addition to traditional fluorescence and absorption sensors, emerging technical directions in this area such as plasmon-enhanced CPs fluorophores [51, 52], waveguide-integrated CPs interferometers [53], and cavity coupled absorption sensors [54] have also been developed.
Related citation in the “References” section:
[51] Liu, W.; Shi, Y.; Yi, Z.; Liu, C.; Wang, F.M; Li, X.L; Lv, J.W; Yang, L.; Chu, P. K. Surface plasmon resonance chemical sensor composed of a microstructured optical fiber for the detection of an ultra-wide refractive index range and gas-liquid pollutants. Opt. Express 2021, 29, 40734–40747. (you suggested)
[52] Nimbekar, A.A.; Bhatia, P.G.; Deshmukh, R. R. Ammonia sensors manufactured by plasma enhanced grafting of conducting polymers on nylon-6 fabrics. Synth. Met. 2021, 279, 116840.
[53] Mumtaz, F.; Zhang, B.H.; Subramaniyam, N.; Roman, M.; Holtmann, P.; Hungund, A.P.; Hungund, A.P.; O’Malley, R.; Spudich, T.M.; Davis, M.; Gerald II, R.E.; Huang, J. Miniature optical fiber Fabry–Perot interferometer based on a Single-Crystal Metal–Organic framework for the detection and quantification of benzene and ethanol at low concentrations in nitrogen gas. ACS Appl. Mater. Interfaces 2024, 16, 13071–13081. (you suggested)
[54] Rossi, S.; Olsson, O.; Chen, S.Z.; Shanker, R.; Banerjee, D.; Dahlin, A.; Jonsson, M.P. Dynamically tuneable reflective structural coloration with electroactive conducting polymer nanocavities. Adv. Mater. 2021, 33, 2105004.
- Long‑term stability and humidity‑induced drift are acknowledged as persistent obstacles, but the manuscript offers only qualitative comments. What are the dominant chemical or mechanical degradation pathways (e.g., dedoping, backbone oxidation, water sorption) and what quantitative lifetime data exist from accelerated‑aging or field trials?
Response: Thank you for highlighting the lack of quantitative detail on long-term stability. In response, we have added a new subsection “3.2 Critical issues in real applications 3.2.1 Long-Term Stability and Humidity Drift: Mechanisms and Quantitative Evaluations ” in the revised version, as shown below.
“3.2.1 Long-Term Stability and Humidity Drift: Mechanisms and Quantitative Evaluations
Although CPs based gas sensors can be used to detect multiple gases and exhibit excellent performances, in practical application scenarios, they still face two key challenges: humidity-induced drift and poor long-term stability. Especially when operating in a variable environment for a long time, these problems seriously restrict their reliable working. The root cause of the performance degradation lies in the chemical, electrochemical, and mechanical changes that occur at the interface between the CPs based sensing layer and the matrix electrode.
In a humid environment, due to the possible cross-reaction between different gases, such sensor may produce similar responsiveness to non-target gases. Therefore, the gas sensing performance may be a mixed response to the target gas and water vapor [141]. On the one hand, water adsorption consumes the adsorption sites of sensing materials, leading to the misjudgment of the concentration of target gases and reducing measurement accuracy. On the other hand, water molecules could interact with polymer chains and disrupt the charge transport path. In addition, the oxidizing or reducing gases may react with the main molecular chain of CPs or their composites, to destroy the conjugated system, reduce the carrier density, and weaken the conductivity and sensing ability of the sensor [142]. For example, in PEDOT:PSS, the PSS shell layer will adsorb water molecules, causing volume expansion, leading to an increase in the distance between PEDOT cores, and thus reducing the carrier transmission efficiency, and further hindering the target gas from reaching the sensing site, prolonging the response time and reducing the sensitivity [143]. By designing a multi-dimensional structure or compounding with inorganic materials, the adsorption sites and charge transfer efficiency of CPs based sensing materials can be effectively enhanced, thereby improving the responsiveness and selectivity of related sensors. For example, inorganic two-dimensional (2D) nanomaterials such as rGO and MXenes have been introduced to improve their electrical conductivity and enhance their mechanical properties [73, 95]; Using other polymers with good hydrophobicity and thermal stability, such as PDMS and fluorinated polyurethane (PU) to encapsulate these sensors [121, 124, 125]
Another important challenge for gas sensors is their long-term stability. This not only exists in CPs-based but also other sensors. This is because during long-term use, sensing active materials may experience performance degradation or even failure due to mechanical external forces, environmental erosion, aging, and other facters. For example, when CPs-based sensors are exposed to air for a long time, due to the obvious dedoping effect, the adsorption sites of CPs may be affected, resulting in a significant decline in sensing performance. At the same time, the presence of oxygen could cause the degradation of CPs, thereby reducing their conductivity [144]. In flexible or wearable devices, mechanical stress cycling may cause the delamination at the interface between the CPs films and the substrates, leading to the interface failure and functional loss, finally affecting the related sensing performances and shortening the service life of such sensors.”
- Wearable and biomedical applications are highlighted, but the review omits any assessment of biocompatibility, toxicology, or regulatory requirements for prolonged skin contact or implantation of CP materials. What data and standards (ISO 10993, FDA class definitions, etc.) support the safe translation of these sensors to human use? Need more clarification.
Response: Thank you for this profound insights. According to your suggestion, we have added a new subsection “3.2.2”, summarizing the biocompatibility assessment, toxicological issues, and regulatory standards related to CPs-based wearable and biomedical sensors, and organized it into a new Table 2.
Please find them as following or in our revised manuscript:
“3.2.2 Biocompatibility, Toxicology, and Regulatory Considerations for CPs-Based Wearable and Biomedical Sensors
CPs, especially PANi, PPy, PThs, PEDOT, and PEDOT:PSS, have been widely explored for wearable and biomedical sensing platforms. However, translating these materials from laboratory prototypes to clinically viable devices necessitates careful consideration of biocompatibility, cytotoxicity, and regulatory compliance under prolonged skin contact or implantation scenarios.
3.2.2.1 Biocompatibility and Toxicological Assessment
PANi, while conductive and easy to process, may release toxic degradation products (e.g., aniline) if not adequately stabilized. Doping agents and processing residues significantly influence its biocompatibility, Composites or coatings (e.g., with biopolymers like chitosan or silk fibroin) are often employed to improve its safety [145, 146]
PPy has demonstrated favorable in vitro and in vivo biocompatibility, so it is generally considered non-cytotoxic, especially when synthesized electrochemically without residual monomers or toxic dopants. Several studies show that PPy-coated scaffolds support cell adhesion and proliferation [147,148].
PThs also have excellent biocompatibility, with few adverse reactions in cell culture and in vivo models. So they can also promote the adhesion and differentiation of neural stem cells and has significant potential in neural regeneration and bioelectronic devices that require long-term tissue integration [149, 150].
PEDOT and PEDOT:PSS, especially when purified or treated to remove excess PSS (e.g., with DMSO or ethylene glycol), exhibit low cytotoxicity and good compatibility with fibroblasts and neuronal cells [151]. As a result, PEDOT:PSS has been used in bioelectronic implants, such as cochlear [152] and neural interfaces [153], supporting its suitability for extended contact with biological tissues.
3.2.2.2 Relevant Standards and Regulatory Framework
ISO 10993 Series [154]: For any device involving skin contact (>30 days) or implantation, they mandate a series of biological evaluations including:
(1) ISO 10993-5 (in vitro cytotoxicity);
(2) ISO 10993-10 (skin irritation and sensitization);
(3) ISO 10993-11 (systemic toxicity);
(4) ISO 10993-6 (implantation effects) if the device is implanted.
FDA Device Classification [155]:
Wearable CPs-based sensors typically fall under Class I or Class II medical devices, depending on the intended use (diagnostic vs. therapeutic) and level of invasiveness. Implantable CPs-based sensors, if developed, would likely be Class III (requiring premarket approval), especially for glucose, neurotransmitter, or gas biomarker detection. Material Risk Assessment: For example, PEDOT:PSS was evaluated by the U.S. FDA as part of neural recording systems, and passed biocompatibility testing in several device submissions (e.g., for cortical electrodes).
3.2.2.3 Design Considerations for Clinical Translation
For clinical translation, there are much serious condierations on the material systems design on CPs-based gas sensors, as shown below.
(1) Encapsulation strategies using biocompatible elastomers (e.g., PDMS, TPU) are essential to isolate CPs from direct tissue exposure while maintaining sensing functionality [125];
(2) Incorporation of bioinert and hydrophobic coatings also helps prevent ion leaching and immune response [155];
(3) Long-term implantation trials (animal models >30 days) are necessary to assess chronic inflammation and fibrotic encapsulation [157].”
Table 2. Some biosafety and regulatory standards on CPs
Conducting Polymer |
Biocompatibility |
Toxicity Concerns |
Regulatory Standard |
Ref. |
PANi |
Variable; can release toxic aniline, modified forms, or composites safer |
Aniline toxicity, residual dopants, and degradation products are concerns |
ISO 10993-5, -10; FDA Class I/11 depending on application |
[145, 146] |
PPy |
Generally good; supports cell adhesion and proliferation |
Low toxicity if properly synthesized |
ISO 10993-5, -10, -11; preclinical animal studies needed for implants |
[147, 148] |
PTh |
General, modification or compounding for improvement |
The toxicity of degradation products is unknown; residual monomers may be toxic |
ISO 10993-5, -10; in some cases, additional in vitro and in vivo testing may be required based on application |
[149, 150] |
PEDOT |
Excellent; used in neural and cardiac interfaces |
Minimal; depends on dopants and processing additives |
ISO 10993-1, -5, -6, -10, -11; used in FDA-cleared implants |
[151] |
PEDOT:PSS |
Good after PSS removal or treatment (DMSO/EG); lowcytotoxicity |
Excess PSS may irritate; removal improves safety |
ISO 10993-5, -10; reviewed under FDA Class 11 submissions (e.g., neural devices) |
[152, 153] |

Reviewer 2 Report
Comments and Suggestions for Authors
This manuscript presents a comprehensive review on the principles, materials, and applications of conducting polymers-based gas sensors. It is well organized, and provides valuable insights for broad researchers in the field. However, there are still some minor issues with the manuscript that should be carefully revised.
- In the keyword section, “conducting polymers” has an inclusion relationship with “polypyrrole”, replace or delete one of them to enhance the representativeness.
- Some terms should be consistent all through the manuscript, such as “conducting polymers” and “conductive polymers”, “gas sensing” and “gas-sensing”, and etc, the authors should carefully check the whole manuscript.
- When mentioning Nernst's equation and Faraday's law, the authors only describe their functions in words. It is recommended to supplement the relevant formulas.
- The authors are encouraged to add a table to summarize the performance of different CPs based gas sensors, including sensing mechanism, target gas, sensitivity.
- It is suggested to add some discussion about the use of AI techniques in the future development of this field.
Author Response
Response to the reviewer 2’s comments
Firstly, we deeply appreciate the reviewer for the hard work on our manuscript. Those comments are all valuable and very helpful for revising and improving our paper. Below you will find our point-by-point response to your comments (Answers are marked in red):
*********************************************************************
Reviewer 2’s comments:
This manuscript presents a comprehensive review on the principles, materials, and applications of conducting polymer-based gas sensors. It is well organized and provides valuable insights for broad researchers in the field. However, there are still some minor issues with the manuscript that should be carefully revised.
- In the keyword section, “conducting polymers” has an inclusion relationship with “polypyrrole”, replace or delete one of them to enhance the representativeness.
Responce: Thank you very much for your careful review and professional suggestions. According to your suggestion, we have deleted "polypyrrole" from the “Keywords” section.
- Some terms should be consistent all through the manuscript, such as “conducting polymers” and “conductive polymers”, “gas sensing” and “gas-sensing”, and etc., the authors should carefully check the whole manuscript.
Response: We have carefully checked the full text and unified the relevant terms into "conducting polymers" and "gas sensing" to ensure the consistency of terminology usage and avoid confusing readers.
- When mentioning Nernst's equation and Faraday's law, the authors only describe their functions in words. It is recommended to supplement the relevant formulas.
Response: Thank you very much for your professional advice. According to your suggestion, we have added the specific formulas of these two laws in the relevant “2.1.1 Electrochemical sensing” section of the revised manuscript” as follows.
“The magnitude of the response follows Nernst's equation ( ) and Faraday's law ( ) providing quantitative information on the gas concentration [25, 26].”
4.The authors are encouraged to add a table to summarize the performance of different CPs based gas sensors, including sensing mechanism, target gas, and sensitivity.
Response: Thank you very much for your valuable suggestions. In the revised manuscript, we have added a new table (Table 1, as shown below) to systematically sort out key information such as the sensing mechanism, target gas, and response time of different CPs-based gas sensors.
Table 1. Comparison of the critical application parameters of different CPs-based sensors for special gas analytes
Mechanism |
Sensor Materials |
Target |
LOD |
Responce |
Ref. |
Chemiresistive |
PEDOT:PSS/PPA |
CO |
50 ppm |
58 s |
[104] |
Chemiresistive |
PANI/SnO2 |
C6H6 |
0.4 ppm |
33 s |
[107] |
Mechanical |
poly(DTCPA-co-BHTBT)-CB |
C7H8 |
4 ppm |
36.47 s |
[48] |
Chemiresistive |
TSP-nAu-PANi |
CHCl3 |
– |
360 s |
[132] |
Chemiresistive |
GO:PEDOT:PSS |
CH3OH |
– |
24.47 s |
[103] |
Optical |
POF/PPy |
C2H5OH |
140 ppm |
5 s |
[61] |
Chemiresistive |
SWCNT/C4F-PPy |
C3H6O |
1 ppm |
750 s |
[111] |
Chemiresistive |
SnO2/PTh |
C3H6O |
0.5 ppm |
10 s |
[35] |
Chemiresistive |
PANi/AgNWs/Silk |
TMA |
1.38 ppm |
90 s |
[24] |
Chemiresistive |
PANi |
H2 |
1 ppm |
15 s/17 s |
[78] |
Chemiresistive |
CA-PANi |
H2S |
1 ppm |
1 s |
[16] |
Chemiresistive |
PANi/MOX |
H2S |
< 100 ppm |
10 s |
[88] |
FET |
Si/PANI:DBSA |
NH3 |
– |
2 s |
[66] |
Chemiresistive |
PEDOT-PVDF |
NH3 |
10 ppm |
80 s |
[99] |
Chemiresistive |
PA-PANI/GO |
NH3 |
25 ppm |
– |
[73] |
Chemiresistive |
f-MWCNT-PEDOT:PSS |
NH3 |
< 10 ppm |
228 s |
[109] |
Chemiresistive |
PPy-TcCoPc |
NH3 |
50 ppm |
8 s |
[84] |
Chemiresistive |
CdS QDs-PTh |
NH3 |
10 ppm |
0.6 s |
[36] |
Electrochemical |
PEDOT:PSS/IrOx Ps |
NH3 |
8 ppm |
87±9 s |
[30] |
Chemiresistive |
PANi |
NH3 |
2.5 ppm |
110 s |
[22] |
Chemiresistive |
PPy@LIG |
NH3 |
1 ppm |
450 s |
[86] |
Chemiresistive |
PANi/FMWCNT |
NH3 |
1 ppm |
15 s |
[77] |
Chemiresistive |
PANi-MWCNTs/PDMS |
NH3 |
10 ppb |
– |
[133] |
Chemiresistive |
RGO-PTh |
NO2 |
0.52 ppm |
498 s |
[95] |
Chemiresistive |
MWCNTs/PANi |
NH3 |
0.3 ppm |
21 s |
[76] |
Piezoelectricv |
GO:PEDOT:PSS |
NO2 |
175 ppb |
35 s |
[43] |
Chemiresistive |
PTh |
NO2 |
0.25 ppm |
4980 s |
[114] |
Chemiresistive |
PANi/BP |
NO2 |
< 2 ppm |
98 s |
[128] |
Chemiresistive |
Au-ZnO-PANi |
NO2 |
< 10 ppm |
600 s |
[129] |
- It is suggested to add some discussion about the use of AI techniques in the future development of this field.
Response: Thank you very much for this valuable suggestion. According to it, we have added a brief discussion about the future development and application of AI technology in the field of CPs-based gas sensing.
In the section of “4 Conclusion and Prospects”:
“In addition, artificial intelligence (AI) technology has great potential in the future development of gas sensors on the basis of CPs. In terms of data processing, AI can efficiently analyze the massive and complex data generated by sensors towards real and unpredictable gas analytes, like E-noses for pneumoconiosis screening and diagnosis [158]. For example, deep learning algorithms could deeply mine the response data of sensors under different gas concentrations and environmental conditions and establish accurate gas recognition models, thereby greatly improving the recognition accuracy of sensors for target gases and effectively reducing misjudgments and omissions. In terms of adaptive adjustment, AI may dynamically adjust the working mode and parameter settings of sensors according to real-time environmental parameters and historical data of sensors. For example, when there are large changes in environmental humidity or temperature, the compensation mechanism of the sensor may be automatically optimized through AI algorithms to keep the sensor in the best detection performance at all times and further improve its stability and reliability in complex and changeable environments. Moreover, with the predictive analysis ability of AI, it is possible to predict in advance possible failures or performance declines of sensors so that timely maintenance and replacement can be carried out to reduce the risk of equipment operation and improve the operating efficiency of the entire gas detection system. By deeply integrating AI technology with CPs based gas sensors, it is expected to promote leapfrog development in this field and create a more intelligent and efficient new era of gas detection.”

Reviewer 3 Report
Comments and Suggestions for Authors This review manuscript on conducting polymers-based gas sensors is structured into several sections: the different transduction modes, the different types of polymers, the nature of the main constituents of the sensor device, and the main applications of these gas sensors. In a "conclusion and prospects" section, several challenges and opportunities for the future of these sensors are discussed. In the applications section, it should be interesting to compare (in a Table) and discuss the different sensors for the detection of the same gas target. Through the comparison, the best solution could appear.Author Response
Response to the reviewer 3’s comments
Firstly, we deeply appreciate the reviewer for the hard work on our manuscript. Those comments are all valuable and very helpful for revising and improving our paper. Below you will find our point-by-point response to your comments (Answers are marked in red):
*********************************************************************
Reviewers 3’s comments:
This review manuscript on conducting polymers-based gas sensors is structured into several sections: the different transduction modes, the different types of polymers, the nature of the main constituents of the sensor device, and the main applications of these gas sensors. In a "conclusion and prospects" section, several challenges and opportunities for the future of these sensors are discussed.
Response: Thank you very much for your positive comment.
In the applications section, it should be interesting to compare (in a Table) and discuss the different sensors for the detection of the same gas target. Through the comparison, the best solution could appear.
Response: Thank you very much for this helpful comment. According to your and another reviewer’s suggestion, we have added a new “Table 1” as shown below in the revised version and gave some brief discussion in the “3 Application of CPs-based gas sensors” section.
Table 1. Comparison of the critical application parameters of different CPs-based sensors for special gas analytes
Mechanism |
Sensor Materials |
Target |
LOD |
Responce |
Ref. |
Chemiresistive |
PEDOT:PSS/PPA |
CO |
50 ppm |
58 s |
[104] |
Chemiresistive |
PANI/SnO2 |
C6H6 |
0.4 ppm |
33 s |
[107] |
Mechanical |
poly(DTCPA-co-BHTBT)-CB |
C7H8 |
4 ppm |
36.47 s |
[48] |
Chemiresistive |
TSP-nAu-PANi |
CHCl3 |
– |
360 s |
[132] |
Chemiresistive |
GO:PEDOT:PSS |
CH3OH |
– |
24.47 s |
[103] |
Optical |
POF/PPy |
C2H5OH |
140 ppm |
5 s |
[61] |
Chemiresistive |
SWCNT/C4F-PPy |
C3H6O |
1 ppm |
750 s |
[111] |
Chemiresistive |
SnO2/PTh |
C3H6O |
0.5 ppm |
10 s |
[35] |
Chemiresistive |
PANi/AgNWs/Silk |
TMA |
1.38 ppm |
90 s |
[24] |
Chemiresistive |
PANi |
H2 |
1 ppm |
15 s/17 s |
[78] |
Chemiresistive |
CA-PANi |
H2S |
1 ppm |
1 s |
[16] |
Chemiresistive |
PANi/MOX |
H2S |
< 100 ppm |
10 s |
[88] |
FET |
Si/PANI:DBSA |
NH3 |
– |
2 s |
[66] |
Chemiresistive |
PEDOT-PVDF |
NH3 |
10 ppm |
80 s |
[99] |
Chemiresistive |
PA-PANI/GO |
NH3 |
25 ppm |
– |
[73] |
Chemiresistive |
f-MWCNT-PEDOT:PSS |
NH3 |
< 10 ppm |
228 s |
[109] |
Chemiresistive |
PPy-TcCoPc |
NH3 |
50 ppm |
8 s |
[84] |
Chemiresistive |
CdS QDs-PTh |
NH3 |
10 ppm |
0.6 s |
[36] |
Electrochemical |
PEDOT:PSS/IrOx Ps |
NH3 |
8 ppm |
87±9 s |
[30] |
Chemiresistive |
PANi |
NH3 |
2.5 ppm |
110 s |
[22] |
Chemiresistive |
PPy@LIG |
NH3 |
1 ppm |
450 s |
[86] |
Chemiresistive |
PANi/FMWCNT |
NH3 |
1 ppm |
15 s |
[77] |
Chemiresistive |
PANi-MWCNTs/PDMS |
NH3 |
10 ppb |
– |
[133] |
Chemiresistive |
RGO-PTh |
NO2 |
0.52 ppm |
498 s |
[95] |
Chemiresistive |
MWCNTs/PANi |
NH3 |
0.3 ppm |
21 s |
[76] |
Piezoelectricv |
GO:PEDOT:PSS |
NO2 |
175 ppb |
35 s |
[43] |
Chemiresistive |
PTh |
NO2 |
0.25 ppm |
4980 s |
[114] |
Chemiresistive |
PANi/BP |
NO2 |
< 2 ppm |
98 s |
[128] |
Chemiresistive |
Au-ZnO-PANi |
NO2 |
< 10 ppm |
600 s |
[129] |

Round 2
Reviewer 3 Report
Comments and Suggestions for Authors
Accept in present form